# Paralytic Shellfish Toxins in Alaskan Butter Clams: Does Cleaning Make Them Safe to Eat?

**DOI:** 10.3390/toxins17060271

**Published:** 2025-05-28

**Authors:** R. Wayne Litaker, Julie A. Matweyou, Steven R. Kibler, D. Ransom Hardison, William C. Holland, Patricia A. Tester

**Affiliations:** 1CSS Inc. Under Contract to the National Oceanic and Atmospheric Administration, National Ocean Service, National Centers for Coastal Ocean Science, Beaufort Laboratory, Beaufort, NC 28516, USA; 2Alaska Sea Grant Marine Advisory Program, Kodiak Seafood and Marine Science Center, University of Alaska Fairbanks, Kodiak, AK 99615, USA; jamatweyou@alaska.edu; 3National Oceanic and Atmospheric Administration, National Ocean Service, National Centers for Coastal Ocean Science, Beaufort, NC 28516, USA; steve.kibler@noaa.gov (S.R.K.); rance.hardison@noaa.gov (D.R.H.); chris.holland@noaa.gov (W.C.H.); 4OceanTester, LLC, Beaufort, NC 28516, USA; ocean.tester@gmail.com

**Keywords:** paralytic shellfish poisoning (PSP), toxicity risk, safe consumption, *Saxidomus gigantea*, subsistence harvesting

## Abstract

Butter clams (*Saxidomus gigantea*) are a staple in the subsistence diets of Alaskan Native communities and are also harvested recreationally. This filter–feeding species can accumulate saxitoxins (STXs), potent neurotoxins produced by late spring and summer blooms of the microalga *Alexandrium catenella*. The consumption of tainted clams can cause paralytic shellfish poisoning (PSP). Traditional beliefs and early reports on the efficacy of removing clam siphons have created the impression that cleaning butter clams by removing certain tissues makes them safe to eat. However, the toxin distribution within clams can vary over time, making the practice of cleaning butter clams unreliable. This study tested the effectiveness of the cleaning methods practiced by harvesters on Kodiak Island, Alaska. Specifically, butter clams were cleaned by removing different tissues to produce samples of “edible” tissues that were tested for STX content. The results were compared to historical data from a study conducted in Southeast Alaska from 1948 to 1949. Using these data, the risk for an average–sized man and woman consuming 200 g of edible tissue was calculated. The results showed that for clams containing >200 µg STX–equivalents 100 g edible tissue^−1^, no cleaning method reduced the concentration of STXs in the remaining tissue below the regulatory limit. Meals containing >900 µg STX–equivalents 100 g edible tissue^−1^ posed a substantial risk of moderate or severe symptoms. No cleaning method assured that untested butter clams are safe to eat.

## 1. Introduction

The butter clam (*Saxidomus gigantea*), found from the Aleutian Islands (Alaska, AK) to mid California, has been a staple in the subsistence diets of Native communities of Alaska and the temperate, northwest coast of the United States and Canada for thousands of years [1]. It is an abundant and valued species with cultural significance. The Southeast Alaskan cultures and Inuit of Qikiqtarjuaq (formerly Broughton Island), among many others in the Pacific Northwest, are reported to have relied on clams as an essential part of their traditional diets [2]. Shell middens, some as old as 9000 years, document that clams, especially butter clams, were abundant food items [3,4]. Clam digging was often much more than a harvesting activity. It was a social event, often associated with potlatches or seasonal ceremonies.

Some communities gathered clams throughout the year, while others focused on clamming during the winter when hunting and fishing were difficult due to intemperate weather. Another reason for collecting clams primarily in winter may have been concern about PSP during the summer. This illness is caused by STXs, potent neurotoxins produced by phytoplankton species, such as *Alexandrium catenella*, that can bloom seasonally in cool temperate waters. STXs accumulate in shellfish, with over a third (35%) of reported paralytic shellfish poisoning (PSP) illnesses in Alaska caused by consuming contaminated butter clams, likely because they may retain STXs for two years or more [5,6]. From 2011 to 2021, the frequency of annual reported PSP cases ranged from 4 to 17, with most instances of less severe cases not being reported. Symptoms typically occur within 30 to 60 min after consuming tainted clams and include numbness spreading from the lips and mouth to the face, neck, and extremities; dizziness; headache; nausea; vomiting; arm and leg weakness; paralysis; respiratory failure; and, in severe cases, death [7].

In the mid–1940s, the vast shellfish resources along the Alaskan coast were recognized and significant efforts were directed toward shellfish utilization. This included the establishment of a Southeast Alaska clam canning industry in 1943 [8]. In 1946, the peak harvest year, STXs were found in shipments of butter clams by the United States Food and Drug Administration. This led to the discontinuation of clam processing in Southeast Alaska. Following this event, extensive sampling was conducted to determine the STX concentrations in butter clams harvested from Southeast Alaskan beaches [9]. These studies revealed that butter clams harvested in any season could be toxic, that toxins were not evenly distributed among clam tissues, and that removing the siphon reduced, but did not eliminate, PSP risk.

This information regarding the varying toxin content in different butter clam tissues was shared with many coastal communities. In some of these communities, it has become common practice to remove the intestinal tract, gills, and black tip of the siphon before consuming the rest of the clam. Over time, a common misconception has arisen among many subsistence and recreational harvesters that by removing these tissues, the remaining parts of the clam are safe to eat [10]. In other cases, clams are not cleaned at all, particularly when they are small, resulting in no reduction in toxin content. Since the State of Alaska does not test recreationally or subsistence–harvested shellfish, some Native communities have established their own testing procedures [11,12]. Only commercially harvested shellfish intended for interstate commerce are regularly tested for STXs [10].

In this study, three different current or historical datasets regarding STX–eq. concentrations in butter clams were examined to more fully understand (1) how removing different clam tissues impacts STX concentrations, (2) the likelihood of consuming a clam capable of causing mild to serious illness, and (3) how meal size and the weight of the consumer may potentially impact the risk of illness. The first dataset consisted of a reanalysis of the tissue–specific STX–eq. concentrations measured in butter clams collected on Kodiak Island, Alaska, from 2015 to 2018 by Kibler et al. [10]. We know that STXs are not evenly distributed throughout each tissue type [10] and these data made it possible to calculate, for the first time, how all the practical ways of removing specific tissues, or combinations of tissues, would impact the STX–eq. concentrations in the remaining portions of the clam to be consumed. For simplicity, we will hereafter refer to any combination of tissues retained for consumption as the “edible” portion of the clam and the discarded tissues as the “non–edible” portion.

The second dataset was obtained by analyzing the concentrations of STX–eq. in butter clams that were prepared using cleaning methods commonly practiced in communities in Kodiak, AK. The clams were dissected into what was considered edible (body and part of the lower siphon) and non–edible (viscera, gills, black tip of the siphon) portions. Traditionally, it has been assumed that the edible portion of the clam was less toxic than the discarded non–edible tissues. The resulting data provided valuable information on methods used by local harvesters that can be shared with other communities.

The third combined dataset consisted of unpublished, historical STX–eq. data from butter clam samples collected in Southeast Alaska from 1948 to 1949 by Chambers and Magnusson [9] and from 1963–1965 by Neal [13]. Both studies collected time series data on STX concentrations in whole clams, while the Chambers and Magnusson study additionally included STX concentration in whole clams minus the siphon. These data provided valuable information including (a) comparative data on the toxin content of butter clams collected in Southeast Alaska compared to the Kodiak region. This comparison is important due to preliminary evidence of toxin content differences in *A. catenella* cells between Kodiak and Southeast AK [14]; (b) changes in toxin concentration from siphon removal that can be directly compared to one of the cleaning methods evaluated using the Kibler et al. dataset [10]; (c) publication of the original data that underpin our understanding of how cleaning may impact toxicity; and (d) information on the extent to which toxin concentrations may decrease in winter, which is relevant for understanding whether cleaning clams may reduce toxicity more in the winter compared to the summer months when toxic *A. catenella* blooms occur.

It should also be noted that the impact of removing various tissues on toxin content in the studies above is based on the concentration of STX–eq. in edible tissue relative to the regulatory safety limit of 80 µg STX 100 g tissue^−1^ for shellfish [15,16]. While the specificity of this limit has been questioned, it is generally considered sufficient for protecting public health [17]. Because this safety limit is quite conservative, it can be difficult for individuals who consume butter clams regularly, due to their nutritional and cultural requirements, to understand intuitively. This is particularly true in cases where private testing has shown that the STX–eq. concentrations are above regulatory levels in butter clams that have been consumed with no apparent ill effects. A more intuitive approach may be to demonstrate the potential risk of PSP toxicity from eating an average sized meal of untested butter clams. To address this issue, we used a recent modeling study by Arnich and Thébault [18] to estimate how the STX–eq. concentrations in the edible tissue and the size of the meal can impact toxicity, as well as how often harvested clams would likely cause significant illness. The resulting findings on the impacts of cleaning butter clams, from the perspectives of both concentration and amount of toxin ingested, are intended for use in developing more effective educational materials for communicating the risk of PSP to Alaskan communities.

## 2. Results

### 2.1. Effects of Removing Various Tissues, or Combination of Tissues, on Toxin Concentrations in Alaskan Butter Clams

The study conducted on butter clams from Kodiak Island between 2015 and 2018 (Kibler et al. [10]) showed that the concentration of STX–eq. 100 g tissue^−1^ varied among different tissue types (viscera, siphon black tip, siphon neck, and body) was variable (see Appendix B). The average weight of whole shucked butter clams was 15.6 ± 2.2 g. Higher STX concentrations were observed in viscera during the month of June, which is consistent with the time when *A. catenella* blooms often occur (Appendix B, panel A). Blooms can vary in intensity among years and locations, resulting in large variations in the amount of STX accumulated by butter clams. The black tip of the siphon consistently retained >100 µg STX–eq. 100 g tissue^−1^, except for batches in May 2017 (Appendix B, panel B). Toxin concentrations in the neck tissue followed a similar pattern to that observed in the black tip (Appendix B, panel C). The body tissues exhibited the highest STX concentrations in June and July, with the exception of 2017 (Appendix B, panel D). Overall, whole–clam STX concentrations mirrored those of the body tissues alone, consistent with the body contributing the largest amount of tissue (Appendix B, panels D, E).

The extent to which the removal of tissues reduced the concentration of STX–eq. in edible tissues differed. This variability was associated with changes in the tissue STX–eq. concentrations in the different tissues over time. When clams actively feed on *A. catenella* blooms, the STX concentration in the gut contents can be very high (>500 µg STX–eq. 100 g tissue^−1^) (Appendix B Figure A1). During and following a bloom, toxins differentially migrate to other tissues. Because of this variable toxin distribution, the overall trend showed that removing tissues generally reduced the overall toxin concentration in the edible tissues relative to the whole clam (e.g., April and May samples; Figure 1), but not always. For instance, in the July samples, more of the STX–eq. load was present in the body tissue instead of the gut and siphon tip. In this case, the average toxin content in the whole clam on a 100 g basis was less than that found in the body alone (Figure 1). A notable finding was that, in the June and July samples where whole–clam toxicities exceeded 200 µg STX–eq. 100 g tissue^−1^, every type of cleaning method failed to reduce STX concentrations below the recommended regulatory level. The only relatively low toxicities seen during a June collection was in 2016, when the butter clams had toxicities similar to those observed in the April and May samples (Figure 1; Appendix A). This may be due to the clams in the June 2016 collection not having been exposed to an *A. catenella* bloom. The overall percent reduction in STX concentration relative to whole clams for the various edible tissues was as follows: (a) whole clam minus gut, 6.8 ± 20.1%; (b) whole clam minus gut and siphon black tip, 7.9 ± 8.5%; (c) whole clam minus entire siphon, 9.6 ± 17.3%; and (d) clam body only, 15.4 ± 17.8%. The data on variations in STX congener composition in individual tissues and how these differences can affect the STX–eq. estimates are discussed in Kibler et al. [10].

### 2.2. Impact of Siphon Removal on Toxin Concentration in Southeast Alaska Butter Clams

The data from Chambers and Magnusson [9] showed that the siphons of butter clams collected from May 1948 to January 1949 in Southeast Alaska had high but variable concentrations of STX–eq. (Figure 2; Appendix A). Although this tissue made up less than 20% of the overall body weight, it substantially contributed to the total toxin content in the clam. Removing this tissue led to a substantial decrease in overall STX content in the remaining tissues (Table 1). Despite this, removing the siphon from butter clams containing >200 µg STX–eq. 100 g tissue^−1^, which was the case for a majority of the clams tested, only rarely reduced the toxin content in the edible tissues below the regulatory level of 80 µg STX–eq. 100 g tissue^−1^ (Figure 2).

### 2.3. Seasonal Changes in Butter Clam STX Concentrations

The Kodiak study conducted by Kibler et al. [10] from 2015 to 2018 did not involve the collection of samples during colder months from November to April. Consequently, the study was unable to assess the effectiveness of different cleaning methods during this time of year. The time series of whole–clam STX concentrations obtained from Chambers and Magnusson [9] and Neal [13] (Figure 3; Appendix A) showed that STX–eq. concentrations declined in some winters but not others. The December and January data (Figure 2 also) showed that high STX–eq. concentrations can persist into the winter months. None of the cleaning methods tested using the Kibler et al. [10] data would reduce STX–eq. concentrations in the higher–toxicity butter clams below the regulatory level for safe consumption.

### 2.4. Comparison of Two Cleaning Methods Practiced by Alaskan Native Communities

Both Method 1 (where the visceral contents are gently squeezed out and the gills and siphon black tip are removed), and Method 2 (where the visceral mass, gills, and siphon black tip are removed) reduced STX–eq. concentrations in the edible tissue on average, but the amount of that reduction varied (Figure 4). When toxin levels in the whole clam exceeded 200 µg STX–eq. 100 g tissue^−1^, neither cleaning method consistently lowered STX concentrations below the regulatory limit of 80 ug STX–eq. 100 g edible tissue^−1^. On average, the reduction in STX–eq. concentrations for Method 1 was 61.4 ± 20.4% (N = 13) relative to whole clams and for Method 2 it was 45.2 ± 23.3% (N = 21) (Appendix A).

### 2.5. Risk Associated with Consuming Meal Contaminated by Saxitoxins

The probability of developing PSP symptoms when a certain amount of STX–eq. is consumed by an average sized man or woman in the United States was calculated based on the data presented by Arnich and Thébault [18], and is shown in Table 2. Using the concentration data from Figure 1, the amount of STX that would be ingested if 200 g of whole or variously cleaned (edible) butter clam tissues were consumed in a single meal was calculated (Figure 5). A comparison of the toxin consumed in a 200 g meal with the amounts of toxin that cause illness showed that most of the butter clams assayed in the 2015–2018 Kodiak contained a sufficient toxin concentration to be of concern (Table 2).

## 3. Discussion

This study demonstrated that no cleaning method for Alaskan butter clams reliably reduced STX concentrations enough to make them safe for consumption (Figure 1 and Figure 2). Whole clams routinely contain between 200 and 500 µg STX–eq. 100 g tissue^−1^ during the summer months, with concentrations between 500 and 1000 µg STX–eq. 100 g tissue^−1^ not being uncommon (Figure 1 and Figure 2). In fact, 63% (N = 447) of the whole–clam batch samples in this study had STX–eq. concentrations exceeding the regulatory limit of 80 µg STX–eq. 100 g tissue^−1^. The combined data, including the cleaning methods practiced in local Kodiak AK communities, also showed that when clams contained >200 µg STX–eq. 100 g tissue^−1^, the cleaning methods rarely produced concentrations in edible tissues below the 80 µg STX–eq. 100 g tissue^−1^ regulatory level (Figure 1, Figure 2 and Figure 4). In some instances where the STX contamination had time to migrate into the body of the clam, the removal of other tissues containing lower STX–eq. concentrations actually resulted in increased toxin concentrations in the ingested tissues. Chambers and Magnusson [9] also reported no correlation between the STX content in butter clams and beach terrain, tidal magnitude, water temperature, or amount of daylight. This reinforces the reality that without testing, it is impossible to predict when a highly toxic clam will be harvested.

Another common practice in Alaska is harvesting butter clams in the winter, which is logical given that *A. catenella* blooms do not occur during that time. The results presented here indicate that STX concentrations in butter clams fluctuate seasonally, but not always to the same extent. In the 1964 study, the concentrations increased in summer, and then declined and remained relatively low throughout the rest of the year (Figure 3B). In contrast, during the 1948–1949 study, the STX concentrations were much higher overall and only dropped slightly after the summer months (Figure 3A). Elevated wintertime concentrations are also evident in Figure 2. It is possible that some of the seasonal differences in STX–eq. concentrations observed were due to whether the local clam populations had converted the STX congeners they retained to lower or higher toxicity forms [10]. The data presented do not allow this possibility to be assessed. Cumulatively, these data are consistent with potentially lower STX concentrations during the winter in some years and not others. Consequently, harvesting in the winter does not reliably reduce the probability of consuming toxic butter clams. This is supported by reports from Canada documenting the occurrence of PSP cases in every month of the year [19].

The data also indicated that, based on Arnich and Thébault [18], an average–sized man or woman consuming even 100 µg STX–eq. would have a ~10% chance of developing mild symptoms and a ~2% chance of developing moderate symptoms (Figure 5; Table 2; [18]). This represented 61.5% of the butter clam batches tested in the Kibler et al. [10] study (N = 39; Figure 5). Mild symptoms include headache, abnormal sensations such as tingling, pricking, numbness, dizziness, vertigo, nausea, and vomiting. Moderate symptoms encompass incoherent speech, involuntary eye movement, rapid pulse, lack of voluntary coordination of muscle movements, shortness of breath, and backache. The modeled response to consuming 900 µg STX–eq. was a ~31% chance of mild symptoms, a 13% chance of moderate symptoms, an ~11% chance of severe symptoms, and a ~0.27% chance of death. Severe symptoms include motor speech disorders, difficulty swallowing, suspension of breathing, weakness of arms and legs, pronounced respiratory difficulties, muscular paralysis, and respiratory arrest (without death). The modeled response based on the reanalyzed data from Kibler et al. [10] indicated that 7.7% of the samples contained >900 µg STX–eq. (N = 39; Figure 5). The findings of this study on the risk of consuming butter clams containing different amounts STX–eq. in an average meal are consistent with the study by Gibbard and Naubert [19], which predicted that consuming 200, 1000, and 2500 µg STX–eq. is needed to cause mild, severe, or extreme symptoms. McFarren et al. [20] also estimated that the ingestion of between 2000 and 4000 µg STX–eq. constitutes a lethal dose.

It important to note that there is a much higher risk of death from severe symptoms in Alaska because many of the communities consuming clams for subsistence are located in remote areas without adequate medical care. Transport times to medical facilities are long, which can exacerbate the severity of symptoms and substantially increase health risks for Alaska residents.

## 4. Conclusions

Given that shellfish are the third most commonly consumed traditional food among Natives in Alaska [21] and that a disproportional number of PSP victims are Alaska Natives (53%) [6], with butter clams accounting for 35% of reported PSP incidents, dissuading recreational and subsistence harvesters from consuming untested shellfish, especially butter clams, is imperative. Our results unequivocally show that no cleaning method reliably makes untested butter clams safe to eat. Furthermore, there is no means of reliably predicting when and where untested, highly toxic clams will be harvested. The findings also indicate that while the probability of severe illness from consuming cleaned clams is not common, it can occur and that unreported mild to moderate illness from consuming cleaned clams is likely a common occurrence.

Involving community members in testing the effectiveness of their traditional shellfish cleaning methods, as was carried out in this study, represents an important step in providing credible information to the population most vulnerable to PSP. The results from this study on cleaning efficacy, the potential risk of consuming a highly toxic clam, and the amount of toxin potentially consumed in an average–sized meal can also be used to develop other outreach and education materials regarding the risks of consuming untested butter clams.

## 5. Materials and Methods

### 5.1. Butter Clam Collection

Batches of 10–15 butter clams were collected for processing. Where possible, three separate batches of butter clams were collected within a two–meter radius. Small specimens (<4 cm in length) were avoided as they were considered too small for subsistence harvesting. The clams were then scrubbed, rinsed with tap water to remove sediment and debris, shucked, and drained [22]. These clams were dissected in various ways to produce different edible and non–edible portions as described in Section 5.3. The µg STX–eq. 100 g tissue^−1^ for each portion was then determined using High–Performance Liquid Chromatography coupled with fluorescence detection (HPLC–FLD).

### 5.2. Saxitoxin Analysis

Because of the high degree of variability in STX–eq. concentrations in individual clams, determinations regarding the safety of butter clams are based on assaying batches of 12 homogenized clams. Concentrations exceeding 80 µg STX–eq. 100 g tissue^−1^ measured via mouse bioassay or HPLC–FLD are considered to exceed the regulatory limit. The mouse bioassay represents a functional assay where time to death is determined and converted to mouse units (MU). One MU is defined as the minimum quantity of toxin needed to kill a mouse within 24 h. HPLC methods measure the structural variants of STX (congeners) found in a sample. Because these congeners vary significantly in toxicity, the total STX–eq. is calculated by summing each measured congener concentration multiplied by its Toxicity Equivalency Factor (TEF) [23]. STX is assigned a TEF of 1.0, with higher toxicity congeners having a TEF > 1.0 and lower ones assigned TEF values < 1.0. One MU is equal to 0.2 STX–eq. [15]. Samples taken as part of this study were analyzed by HPLC and any historical data containing concentrations in MU 100 g tissue^−1^ were converted to µg STX–eq. 100 g tissue^−1^ as described above.

All tissue samples for HPLC–FLD analysis were immediately frozen at −20 °C and held pending shipment to the National Oceanic and Atmospheric Administration (NOAA) Laboratory in Beaufort, North Carolina, for analysis. These samples were analyzed via HPLC with precolumn oxidation using the standard methods of Lawrence et al. [24], refined by Ben–Gigirey et al. [25] and Harwood et al. [26]. Briefly, the samples were processed using a Kinematica Polytron model PT–MR 2500E homogenizer fitted with a 1 2 mm dispersing head (Kinematica, Inc., New York, NY, USA). A 5 g subsample of homogenized tissue was extracted with 3 mL 1% acetic acid in a 100 °C water bath for 5 min. After cooling at 4 °C, the sample was centrifuged at 4500 rpm (2940× *g*) for 10 min, and the supernatant was collected. The remaining pellet was re–extracted and the supernatants combined. Then, 1 mL of the combined extract was passed through a conditioned SPE C18 cartridge, pH–adjusted to 6.5, and diluted to 4 mL for oxidation with periodate and peroxide. PSP toxins were quantified using Agilent 1100 (Santa Clara, CA, USA) or Waters Aquity Arc (Milford, MA, USA) HPLC systems coupled to FLD and 5 µm C18 columns (150 × 4.6 mm, Phenomenex, Inc., Torrance, CA, USA). The Agilent fluorescence detector was an Agilent 1100 G1321A FLD and the Waters detector was a 2475 FLR detector. Fluorescence excitation was set to 340 nm and emission to 395 nm. The mobile phase (A) consisted of 0.1 M ammonium formate, adjusted to pH 6 ± 0.1 with 0.1 M acetic acid, while mobile phase (B) consisted of 0.1 M ammonium formate with 5% acetonitrile, also adjusted to pH 6 ± 0.1 with 0.1 M acetic acid. The mobile phase was delivered by an Agilent 1200 series LC. The flow rate was set at 2 mL min^−1^. The LC gradient was as follows: 0–5% mobile phase B in the first 5 min, 5–70% B for the next 4 min, held at 70% B for 1 min, and back to 100% A over the next 2 min. The 100% A was held for an additional 2 min to allow for column equilibration before subsequent sample injections [27]. Concentrations of STX, neosaxitoxin (neoSTX), decarbamoyl saxitoxin (dcSTX), gonyautoxins 2 and 3 (GTX2,3), decarbamoyl gonyautoxins 2 and 3 (dcGTX2,3), gonyautoxins 1 and 4 (GTX1,4), gonyautoxin 5 (GTX5), and the di–sulfated toxins C1 and C2 were quantified using standards purchased from the National Research Council Canada (Halifax, NS, Canada). Isomers GTX 1,4, GTX 2,3, and C1,C2 were not separated using precolumn oxidation and were measured simultaneously. In keeping with the Alaska Department of Environmental Conservation’s protocols, toxicity equivalency factors (TEFs) from the European Food Safety Authority 2009 [16] were used to convert congener concentrations to STX–eq., with the higher TEF used for unresolved congener pairs. Throughout this study, toxin concentrations in clams and tissue components are reported in total STX–eq. The contribution of individual congeners to the clam toxin pool was calculated by weight based on STX–eq. The fraction of clam toxin concentrations associated with specific tissues (% toxin) was calculated as the STX–eq. in each tissue relative to the total toxin pool in that tissue component. The LOQ data for this method are reported in Turner et al. [27].

### 5.3. Determining STX–eq Concentrations in Edible and Non–Edible Tissues

The effects of cleaning on STX–eq. concentrations in edible butter clam tissues were determined in three separate studies, as noted in the Introduction. The various tissue types (viscera, siphon black tip, neck of the siphon, and the main body of the clam) analyzed in the Kibler et al. study [10] are shown Figure 6 for reference. The weights of each tissue type and the corresponding STX–eq. concentrations were used to calculate the concentration in the edible and non–edible portions shown in Table 3. Specifically, the resulting µg STX–eq. per weight of edible tissues was converted to a 100 g tissue basis by multiplying by 100 and dividing by the weight of the edible tissue. The concentrations of STX–eq. in each tissue type for each batch of clams analyzed were then plotted in order of the month of the year in which they were collected. The percent reduction in STX–eq. in edible tissue versus whole clams was calculated as the concentration in whole clams minus that in edible tissue divided by the whole clam concentration and multiplied by 100.

The STX concentration data presented in a 1954 report by Chambers and Magnusson [9] for whole clams and clams without the siphon were assessed using mouse bioassay and converted to STX–eq. as previously described [15,28]. The original data sheets did not specify the weights of the siphon compared to the rest of the body, but the final report indicated that siphons accounted for 14.3 to 19.9% of the body weight, with values of around 18 to 19% being more common. Based on this information, it was assumed that the weight of the non–edible siphons was 18.5% of the total weight and that the remaining 81.5% was edible tissue. Using the µg STX–eq. 100 g tissue^−1^ concentrations in the edible and non–edible tissues, and assuming 100 g of tissue was used for the analysis of STX, the concentration in whole butter clams was calculated. The concentrations of STX–eq. in the whole clams, edible, and non–edible tissues were then plotted by the month of the year in which the batch samples were collected. The percent reduction in STX–eq. in edible tissue compared to whole clams was calculated as described above.

The study of butter clam cleaning practices in communities on Kodiak Island, AK, was conducted as follows. In both Methods 1 and 2, the clams were opened by cutting the adductor muscles, removed whole from the shell, and strained. In Method 1, performed by Alaska Sea Grant personnel, the visceral contents were gently squeezed out, and the gills and black tip of the siphon were removed. The remaining light–colored meat, including the buttons, neck, mantle, and visceral mass with the foot, represented the edible tissue. In Method 2, carried out by a local harvester, the visceral mass (including the gills) was entirely cut out along with the black tip (Figure 7). The remaining light–colored meat, including the buttons, neck, mantle, and foot, represented the edible tissues. For the analysis, three replicate batches containing 12 butter clams each were harvested each month from either Mission Beach (57.7922, −152.3863) between April and July of 2018 or Shipwreck Beach (57.2106, −153.2961) between April and October 2018. The edible and non–edible tissues were then frozen, shipped, and analyzed as described above. The concentrations for whole clams, edible, and non–edible tissues from each batch were again plotted by the month in which the samples were collected, and the percent reduction in STX–eq. concentrations was calculated.

### 5.4. Seasonal Changes in Saxitoxin Concentrations in Butter Clams

The studies examining how different cleaning techniques could affect butter clam STX–eq. concentrations were not conducted during the colder months. If STX–eq. concentrations remained high in clams during the winter, the impacts of cleaning should be similar to those observed during the warmer months. To address this issue, time series data on whole butter clam STX–eq. collected in Southeast Alaska from February 1948 to September 1949 [9] and from January 1963 to March 1965 [13] were utilized. If toxin levels remain comparable to those in summer, then the cleaning results would likely be comparable. The data from the two different time series were plotted separately to illustrate the seasonal variation in toxin concentrations.

### 5.5. Estimates of PSP Risk Based on the Amount of Toxin Consumed in an Average–Sized Meal

The risk associated with an average–weight man (90.8 kg) or woman (77.6 kg) consuming a typical meal containing 200 g of butter clam tissue [18,29] was determined based on the STX–eq. in the edible tissues obtained from the 2015 to 2018 Kodiak study and the modeled relationship between the amount of toxin consumed per body weight and the likelihood of developing different severity symptoms. To calculate this risk, the average body weight for a man and a woman was multiplied by the modeled µg STX–eq kg.b.w.^−1^ for different risk levels [18,29]. This provided an estimate of the total STX–equivalents that would produce no symptoms, mild symptoms, medium symptoms, severe symptoms, or death. Next, we determined the average serving size for butter clams, which is approximately 240 g [30], and erred on the conservative side by assuming a portion size of 200g of edible butter clam tissues per meal. The average portion size was then multiplied by the STX–eq. for edible tissues obtained from the analysis of the Kibler et al. data [10] (Figure 1) to estimate the amount of toxin consumed. The toxicity risk was then assessed by comparing the STX–eq. consumed versus the total STX needed to produce symptoms of different severity.

## Figures and Tables

**Figure 1 toxins-17-00271-f001:**
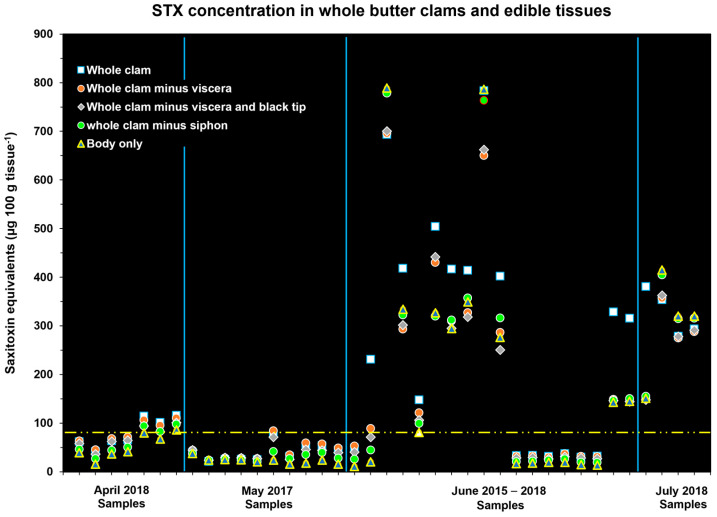
The µg STX–equivalents 100 g tissue^−1^ for whole butter clams and the different types of edible tissues. The yellow dashed line represents the regulatory limit (80 µg STX–eq. 100 g tissue^−1^, N = 39). The blue lines indicate the month of the year in which samples were collected.

**Figure 2 toxins-17-00271-f002:**
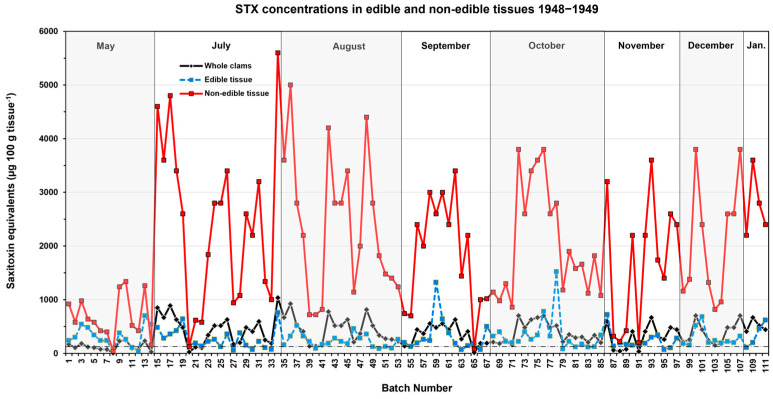
Time series of STX–eq. concentrations in whole butter clams, as well as edible (body and gut) and non–edible (entire siphon) tissues collected in Southeast Alaska from 1948–1949 [9]. The black dashed line represents the regulatory limit (80 µg STX–eq. 100 g tissue^−1^).

**Figure 3 toxins-17-00271-f003:**
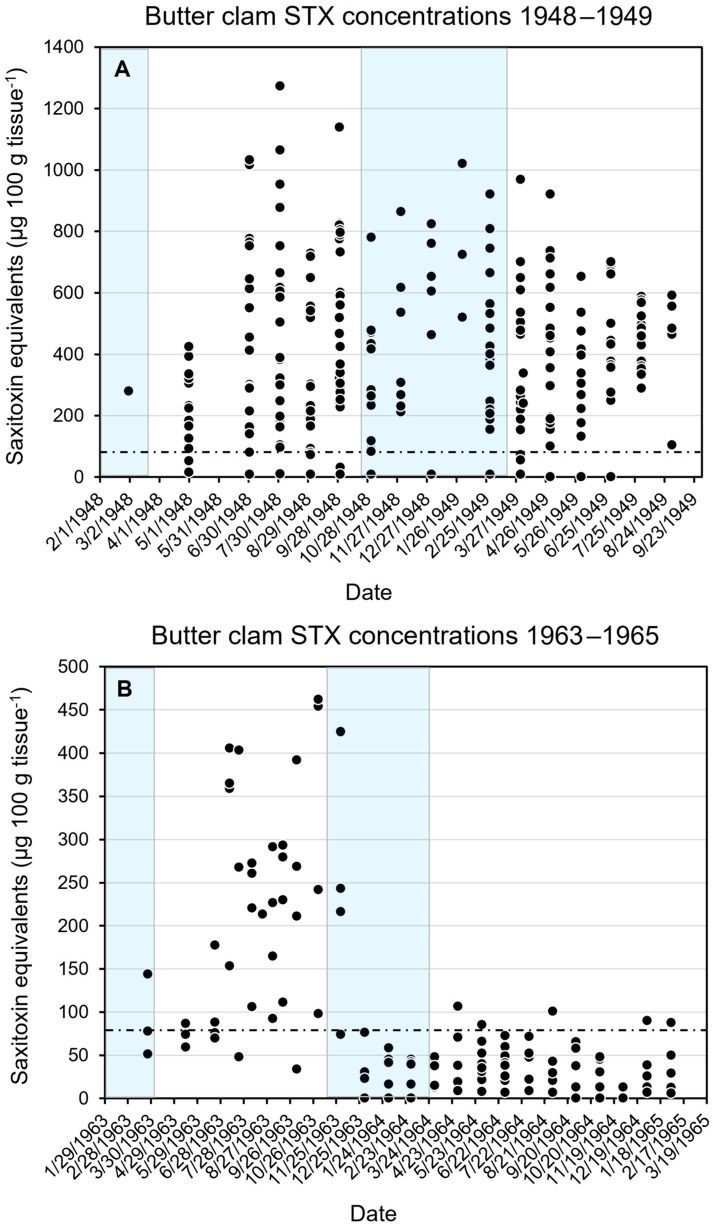
Time series of total clam STX concentration data reported by (**A**) Chambers and Magnusson (1948–1949; N = 250) [9] and (**B**) by Neal (1963–1965; N = 123) [13]. The blue shading on the graphs indicates the period from November to March when water temperatures are cooler. The black dashed line indicates the regulatory limit (80 µg STX–eq. 100 g tissue^−1^).

**Figure 4 toxins-17-00271-f004:**
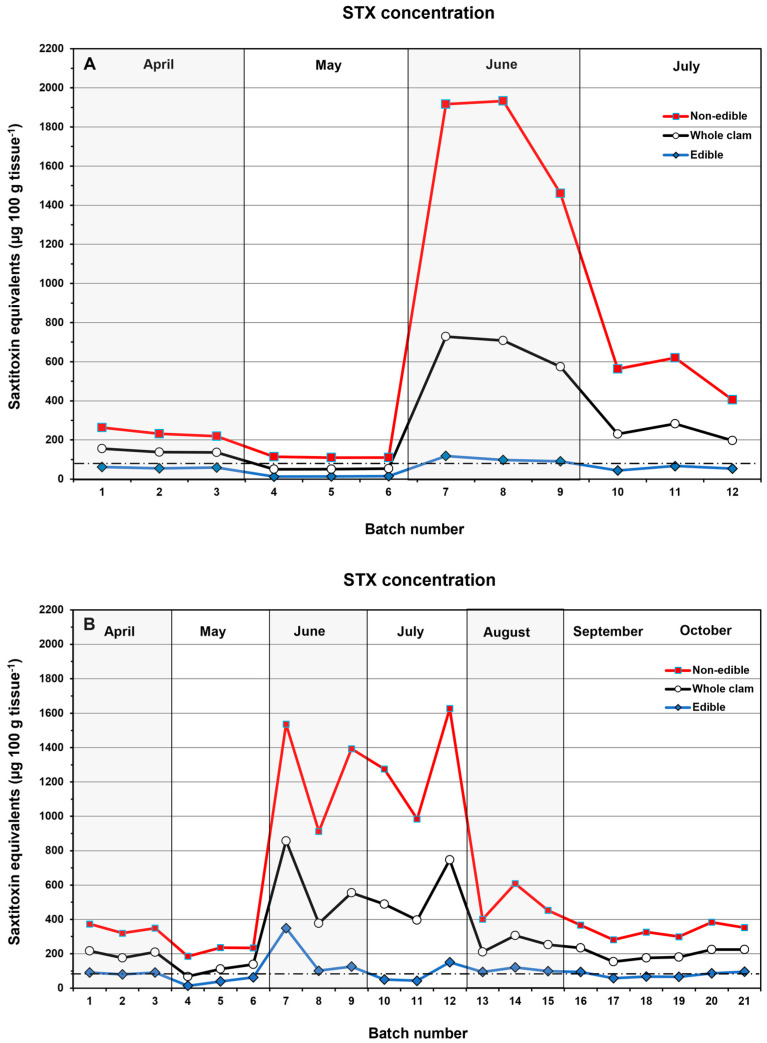
Time series of the STX–eq. concentrations in the edible and non–edible butter clam tissues prepared following (**A**) Method 1 and (**B**) Method 2. Each month, three replicate batches were collected on the same day of the month (April to July 2018 for Method 1, April to October 2018 for Method 2). The three individual replicates collected each month are plotted separately to show inter–batch variability. The black dashed line indicates the regulatory limit (80 µg STX–eq. 100 g tissue^−1^; N = 21).

**Figure 5 toxins-17-00271-f005:**
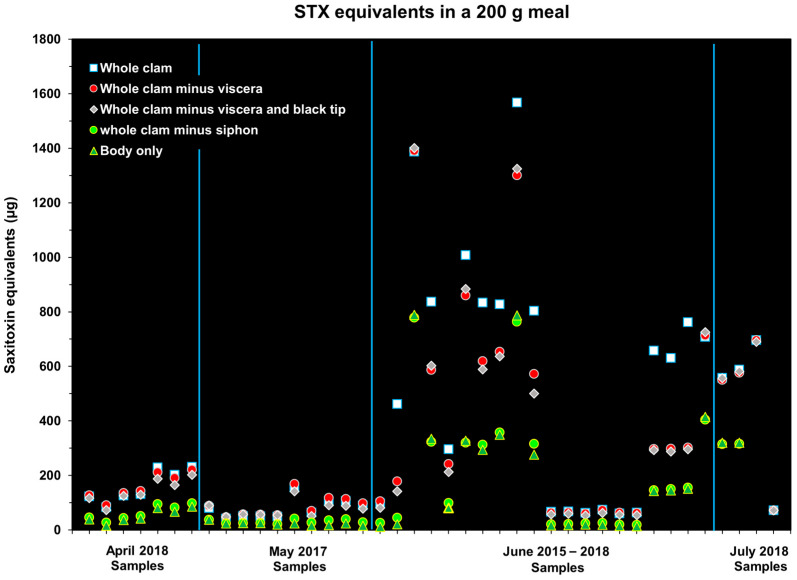
STX–eq. concentrations that would be ingested if 200 g of edible tissue(s) were consumed. The concentration data used to determine the quantity of STX ingested were from the 2015–2018 Kodiak study (see Figure 1, N = 39). The blue lines indicate the month of the year in which samples were collected.

**Figure 6 toxins-17-00271-f006:**
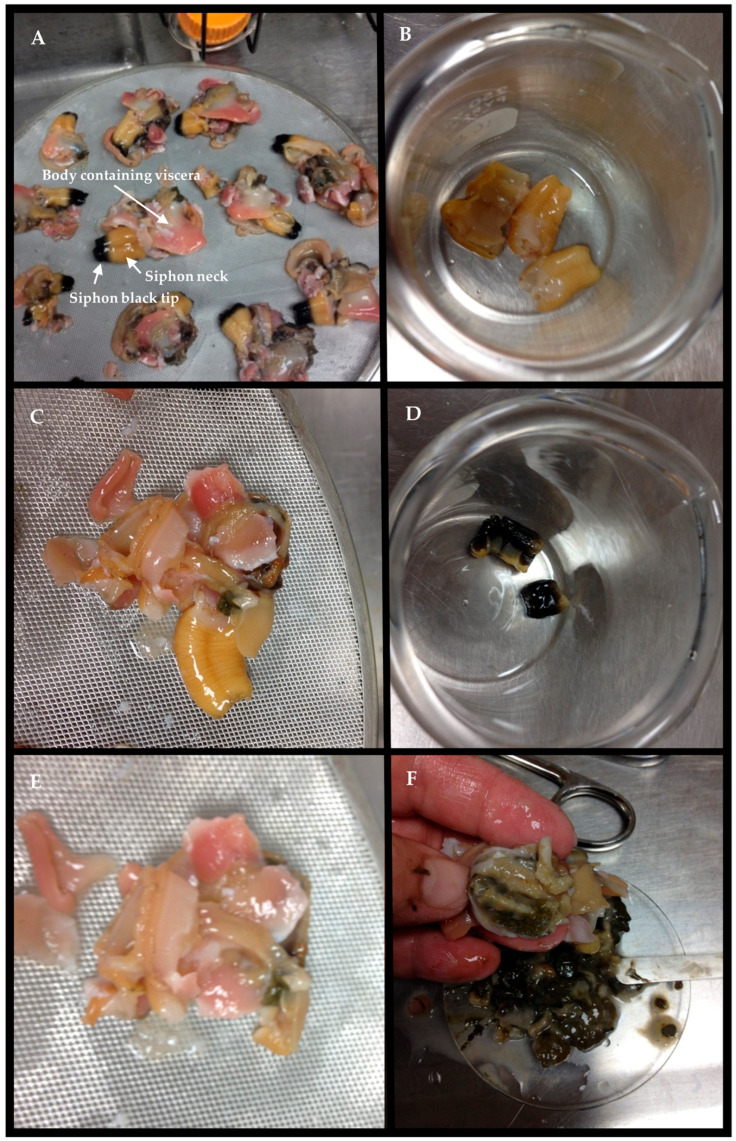
The various dissected clam tissues analyzed for STX concentration. (**A**) Shucked butter clam, (**B**) excised siphons without the black tip, (**C**) body with siphon neck, (**D**) black tips, (**E**) body tissue, (**F**) dissected visceral tissue (viscera shown in dish), and the location of the viscera in the body of the butter clam.

**Figure 7 toxins-17-00271-f007:**
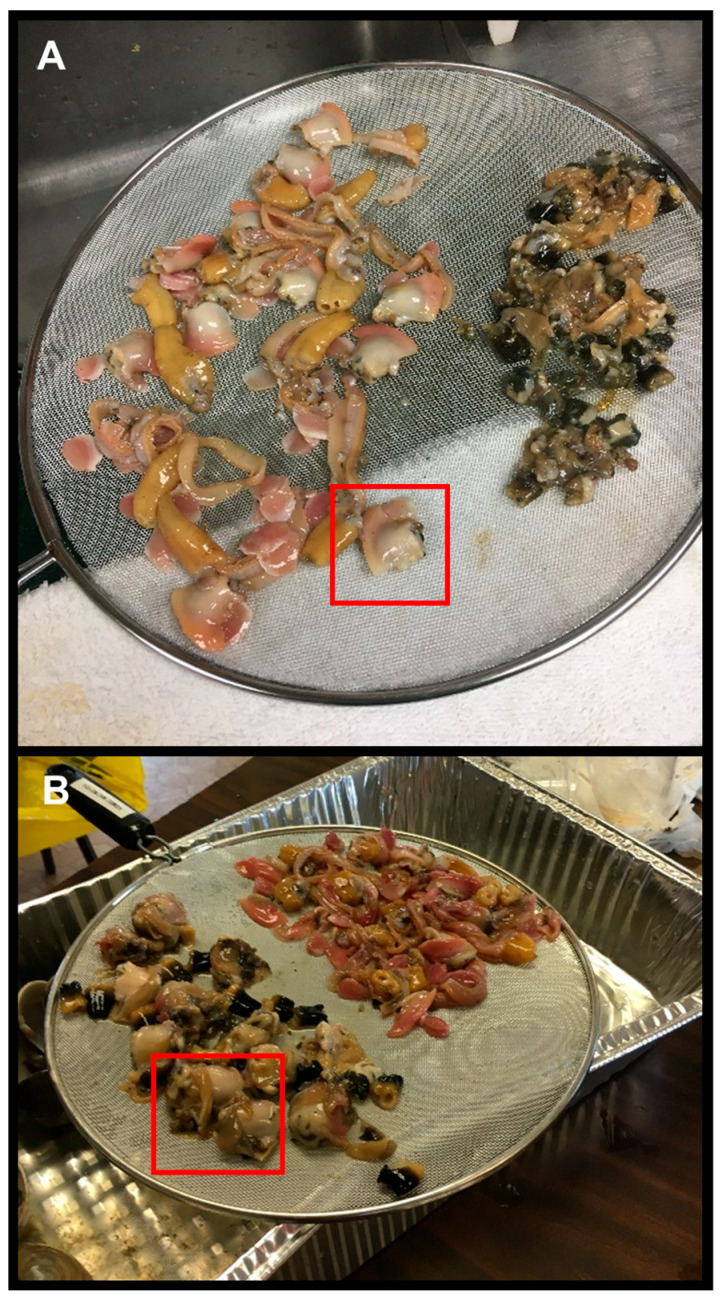
Edible tissues (lighter) versus non–edible tissues (darker) prepared using two methods. (**A**) In Method 1, the visceral content was expressed while keeping the overall visceral mass intact (red square) and the gills and black tip of the siphon was excised. (**B**) In Method 2, both the whole visceral mass including gills (red square) and black tip were removed resulting in less remaining tissue compared to Method 1.

**Table 1 toxins-17-00271-t001:** The percentage reduction in overall STX–eq. concentrations in whole clams compared to the edible tissues calculated from the information presented in Figure 2.

Month	Mean	SD	n
May	22	20	15
July	57	23	19
August	60	18	19
September	48	27	14
October	53	18	17
November	45	28	13
December	52	14	10
January	59	21	4
Overall	49	24	111

**Table 2 toxins-17-00271-t002:** The probabilities of an average–sized man or woman experiencing no symptoms, mild symptoms, severe symptoms, or death after ingesting different amounts of STX–eq. are derived from the toxin kg^−1^ data presented in Arnich and Thébault [18].

Average–Sized Man (90.6 kg) (µg STX–eq. Ingested)	Average–Sized Woman (78 kg) (µg STX–eq. Ingested)	Probability of No Symptoms	Probability of Mild Symptoms	Probability of Moderate Symptoms	Probability of Severe Symptoms	Probability of Death
91	78	88.67%	9.22%	1.57%	0.53%	0.00%
906	775	45.49%	30.66%	12.99%	10.59%	0.27%
9060	7750	7.55%	19.50%	19.43%	46.37%	7.15%
90,600	77,500	0.30%	2.40%	5.30%	47.70%	44.30%
906,000	775,000	0.00%	0.05%	0.26%	11.59%	88.10%

**Table 3 toxins-17-00271-t003:** Different ways in which tissues can reasonably be partitioned into retained edible tissue(s) and discarded non–edible tissue(s).

Tissues Retained as Edible	Tissues Discarded as Non–Edible
Whole clams (includes viscera, black tip at the end of the siphon, siphon neck, and body of the clam)	
Black tip, neck, body	Viscera
Siphon neck, clam body	Viscera, black tip
Clam body	Viscera, black tip, siphon neck
Viscera, clam body	Black tip, siphon neck

## Data Availability

The original contributions presented in this study are included in this article and Appendix A. Further inquiries can be directed to the corresponding author.

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
