# Peer review of "Paralytic Shellfish Toxins in Alaskan Butter Clams: Does Cleaning Make Them Safe to Eat?"

_toxins, 2025, doi:10.3390/toxins17060271_

Round 1

Reviewer 1 Report

Comments and Suggestions for Authors

Comments on the Quality of English Language

The English was OK. 

Author Response

Saxitoxin in Alaskan butter clams: Does cleaning make them safe to eat?

Comment 1. I have read the paper “Saxitoxin in Alaskan butter clams: Does cleaning make them safe to eat?” and it is clear that PSP is a big problem in Alaska and that any information on reducing the risk is useful.

However, I do have some concerns regarding the paper as detailed below.

My major comments are:

What is original data and what has been presented elsewhere?   I got confused when reading the paper over what was determined in this study and what was already available. What was novel?

Response 1. We have sorted out the three data sources used in this study more clearly. These include:

  • Data from reports not published in primary literature (e.g.) Chambers, J. S.; Magnusson, H. W. Seasonal Variations in Toxicity of Butter Clams from Selected Alaskan Beaches; 1950. https://spo.nmfs.noaa.gov/sites/default/files/legacy-pdfs/SSRF53.pdf.

Neal, R. A. Fluctuations in the levels of paralytic shellfish toxin in four species of lamellibranch mollusks near Ketchikan, Alaska, 1963-1965. Dissertation University of Washington, Seattle, 149 pp. 1967. DOI: https://www.proquest.com/openview/7125c2f368b054c09f7110aff55ab1f2/1?pq-origsite=gscholar&cbl=18750&diss=y.

  • Comparative information from Kibler, S. R.; Litaker, R. W.; Matweyou, J. A.; Hardison, D. R.; Wright, B. A.; Tester, P. A. Paralytic shellfish poisoning toxins in butter clams (Saxidomus gigantea) from the Kodiak Archipelago, Alaska. Harmful Algae 2022, 111, 102165. DOI: 10.1016/j.hal.2021.102165
  • New data from this study that include cleaning techniques used by members of Native Kodiak communities and the results of toxicity testing on the tissues they considered safe to eat.

Comment 2.  In the results section 2.1 the study of Kibler et al 2022 has been discussed but this is not really appropriate in the results section as it is from previous work.  Looking at the Kibler paper they discuss distribution of STX in butter clams as well as seasonality – they also talk about whether cleaning makes clams safe to eat.  In section 2.2 of the results it is a different set of data which is discussed (Chambers and Magnusson) so again not attributable to your study. From what I can see only sections 2.2.3 and 2.2.4 are from the current work.

I think in the results section you need your analyses and then in the discussion you can do the comparisons to support your conclusion.

Response 2. We take your point and have made an effort to separate the results of this study from the other two data sets.

Comment 3. Perhaps writing a review type paper would be better?

Response 3. The data streams utilized in this paper represent unpublished data, reanalyzed data and new data. The latter have not been previously published in the primary literature. The early studies from Chambers & Magnusson left the impression in some of the harvesters that cleaning certain Butter Clam tissues would render them safe to eat. The recent study by Kibler et al. tried to remove many of the doubts about what was happening when different Butter Clam tissues were removed. The current study goes a step further and reports the results of studies when Butter Clams are cleaned in traditional ways by local Kodiak Native community members. Native Alaskans more likely to suffer from paralytic shellfish poisoning because shellfish are consumed by 41% of the population ( PMCID:P<C6834783 PMID: 31723724).

The goal of this study was to bring as many relevant sources of information that used the same metrics together as possible to address the question of whether or not cleaning would make clams safe to eat. In this respect we argue this manuscript still fits better as a research article rather than a review article. There is a new facet to the information presented in this study and it comes from the grass roots level of community involvement. This was done with the hope that the results would be understood and accepted by those who need this information the most.

Comment 4. Introduction

The first part of the introduction does not flow well and is repetitive. Line 29 talks about native communities but then this is repeated on line 32 and line 38.  It would be better to link these different bits together.  Similarly, on line 31 abundance is mentioned and then again on line 35.

Response 4. We thought the historical information was interesting, but the reviewer did not agree. To address their concerns we deleted lines 37-39.

“Some villages used traditional cultivation techniques to expand clam habitat by building intertidal terraces called “clam gardens” 3. In other locations like Copper River, plentiful clam beds were available to Native communities 5.”

Comment 5. On line 68 you mention testing strategies but not what they are.

Response 5. To address the confusion, we changed the word strategies for protocols.

Comment 6. What is the incidence of PSP in Alaska.  This is needed to give relevance to your work.

Response 6. We added the requested information. Between 1993-2021 the State of Alaska’s Epidemiology Section received 132 reports of PSP cases from 79 incidents. Of the 85 patients for whom race was recorded 53% were Alaska Natives.

Comment 7. On line 69 you say you use the data of Kibbler then on line 76, here we report so I presume that this bit is the analysis that you did yourself?  If so then this is the important bit and you collected data from elsewhere to compare and help come to a conclusion.

Response 7. We have rewritten this paragraph to better describe the data sources used in this study and how they are relevant to addressing the central question of how effective the different cleaning methods are at reducing toxicity. The pararagraph now reads:

This study utilized three different data sources, all with the goal of examining how removing different butter clam tissues impacted the toxicity of the tissues actually consumed. The first data set included the tissue-specific STX toxicity data from butter clams collected on Kodiak Island, Alaska by Kibler et al. 11 from 2015 to 2018. These data were reanalyzed in greater detail a more detailed analysis of the changes in toxicity with removal of specific tissues than was previously presented. The second data set came from analysis of the STX concentrations in the edible and non-edible portions of butter clams prepared using butter clam cleaning methods practiced by Kodiak AK community members. These data are valuable because they provide specific information on methods used by local harvesters in the region and that can be reported as reliable information to other communities. The third STX data set included unpublished studies done on Kodiak Island from Southeast Alaska in 1948 and 1949. In that study, the whole siphon was removed from the clam and the toxicity of the siphon tissue versus the rest of the clam measured 9. These data offer insight into how this method of cleaning affects the overall toxicity of the clams, plus provides information for butter clams collected in southeast Alaska versus the Kodiak region. The results can also be compared directly to the reanalyzed data from Kibler et al. 11 study. This cross-regional comparison is important because there is evidence differences in the toxicity of shellfish harvested in Kodiak versus Southeast AK 14. Also, since saxitoxin producing A. catenella blooms occur in the spring – summer seasons, this project also used whole clam toxicity data available from 1948-1949 9 and 1963-1965 10 to examine how toxicity changes seasonally due to depuration and how this might impact the efficacy of cleaning to reduce toxicity.

Comment 8. I don’t think the bit line 71-74 is required in the introduction as it is very specific.  I think the intro has too much detail.

Response 8. The section “Specifically, edible tissues were variously defined as whole butter clams, whole clams minus the viscera, whole clams minus the viscera and black tip of the siphon, the whole clam and viscera minus entire siphon (neck and black tip) and the butter clam body alone.” was removed to address the concern regarding excess detail in the introduction.

Comment 9. Line 90. Regulatory limit. I think that the statement that it is based on remnant meals and a safety factor is incorrect.  However, if you have a reference then it should be given. There is debate over the PST regulatory limit so it could be helpful to look at the EFSA, saxitoxin scientific opinion and also some more recent work (Finch, Boente-Juncal) which talk about the appropriateness of the limit.

Response 9. reference describing how the initial level was set has been included as requested along with the information on the more recent work suggested by the reviewer. The first portion of the paragraph now reads as follows.

It should be noted that the impact of removing various tissues on toxicity risk in the above studies is based on the concentration of STX-eq. in edible tissue relative to the regulatory safety limit of 80 µg STX 100 g tissue-1 for shellfish [17]. This regulation is based on the lowest concentrations of STX found in remnant meals that have been shown to cause illness divided by a safety factor of 10 [18]. Though this limit has been questioned, it is generally considered fit for the purposes of protecting public health [19]. While this approach ensures safety, it is quite conservative and may not be informative for individuals who consume butter clams regularly due to their nutritional and cultural requirements. Similar to the belief that cleaned clams are safe to eat, harvesters who have consumed shellfish previously that tested above the regulatory limit with no apparent ill effects, may perceive PST-positive shellfish as posing little risk. Furthermore, toxicity estimates based on shellfish tissue concentrations do not provide an estimate of how exposure will change based on the amount of clam tissue consumed and the weight of the individual. To address this issue, we utilized a recent modeling study by Arnich and Thébault [20] to estimate how the STX-eq. concentrations in the edible tissue and the size of the meal can impact toxicity, as well as how often harvested clams would likely cause significant illness. The resulting findings on the impacts of cleaning butter clams, from both a concentration-based and amount of toxin ingested perspective, are intended for use in developing more effective educational materials for communicating the risk of PSP to Alaskan communities.

The following information was omitted based on the reviewer’s suggestion that there was too much methodological detail provided in in introduction.

Specifically, the Thébault et al. 16 study used data collected from 13 studies they calculated the risk of no, low, moderate and high toxicity or even death based on the quantity of STX-equivalents consumed per kg body weight. To relate these results to the Kibler et al. 11, study, we calculated the STX-eq. that would be ingested if 200 g of edible tissue (the recommended portion size for butter clams is 240 g). We then compared the likelihood of illness if that amount of toxin had been consumed by an average sized man or woman in the United States. This was done to provide harvesters with a better understanding of the risk associated with eating a standard sized meal of untested butter clams.

Comment 10. Line 97. The modelling study was Arnich and Thebault

Response 10. Good catch, the reference has been corrected. Endnote issue.

Comment 11. Line 108. SP or PSP?

Response 11. PSP – the typo has been corrected.

Comment 12. Results

Fig 1 and Fig 2.  Why are the concentrations so much higher in the earlier data?

Response12.  Bloom intensity varies from year to year as does the amount of toxin incorporated. This largely accounts for why the toxicity estimates were so different in the two datasets. The following sentence has been added to address this issue. “These blooms can vary in intensity among years and locations, leading to large variations in the amount of STX accumulated by butter clams.”  Section 2.1 page 4.

Comment 13. Line 197 A brief description should be given of the meaning of method 1 and 2 here and in the figure caption.

Response The requested information was provided in the first sentence of section 2.2.3. The sentence now reads. “Both Methods 1 (the viscera contents are gently squeezed out and the gills and siphon black tip removed) and 2 (the visceral mass, gills, and siphon black tip removed) on average reduced toxicity in the edible tissue, but the amount of reduction was variable (Fig. 4).”

Comment 14. Discussion

Line 261. The data also indicated that an average sized man …  Your data did not show this, 100 ug would give a response of .. 900 ug would give a response of.  This is toxicity data or from the modelling study.  What you did do was relate the amounts in butter clams to this model.

Response 14. The reviewer is correct. In response, we have changed the paragraph as follows: The data also indicated that based on the Arnich & Thébault. [20] an average sized man or woman consuming even 100 µg STX-eq. would have a ~10% chance of developing mild symptoms and a ~2% chance of developing moderate symptoms (Fig. 5; Table 2; [20]). This represented 61.5% of the butter clam batches tested in the Kibler et al. [10] study (N=39; Fig. 5). Mild symptoms include headache, abnormal sensations such as tingling, pricking, numbness, dizziness, vertigo, nausea, and vomiting. Moderate symptoms encompass incoherent speech, involuntary eye movement, rapid pulse, lack of voluntary coordination of muscle movements, shortness of breath and backache. The modeled response to consuming 900 µg STX-eq. was a ~31% chance of mild symptoms, a 13% chance of causing moderate symptoms, an 11% chance of severe symptoms and a ~0.27% chance of death. Severe symptoms include motor speech disorders, difficulty swallowing, suspension of breathing, weakness of arms and legs, pronounced respiratory difficulties, muscular paralysis and respiratory arrest (without death). In the Kibler et al. [10] study, the modeled response indicates 7.7% of the samples contained >900 µg STX-eq. (N=39; Fig. 5). It should be noted that there is a much higher risk of death from severe symptoms because many of the communities consuming clams for subsistence are located in remote areas without adequate medical care, and requiring long delays during transport. This may exacerbate the severity of symptoms and substantially increase health risks in Alaska residents.

Comment 15. Line 276. You did not determine risk from consuming STX as this again is toxicity.  You talk about the risk of eating butter clams.

Response 15. We agree with the reviewer. The changes to this comment are shown in the response to reviewers’ previous comment regarding line 261 above

Comment 16. Methods

Line 344.  You have included methods for things that you did not do.  Having some details of the study is useful but the methods are for what you did.

Response 16. We deleted the information not specific to our work in this study.

Reviewer 2 Report

Comments and Suggestions for Authors

The manuscript describes the efficacy of different cleaning methods applied to butter clams, an important resource for diet of Alaskan Native communities and recreationally activities. These bivalves in fact can accumulate toxins such as saxitoxin causing Paralytic Shellfish Poisoning (PSP). The results obtained, also compared with historical data, showed that the removal of certain tissues not makes clams safe to eat. Moreover the risk associated to the consumption of 200 g of edible tissue for an average sized man and woman was evaluated, underlining that high levels of PSP contamination pose a substantial risk with moderate or severe symptoms.

The manuscript is not clear as regards the experimental results obtained, then it is also difficult to understand the novelty of the research.

Moreover the manner in which the paper is structured seems more a review than an  article. Then a substantial revision should be done to underline the original features of the study.

The” Introduction” should be revised, some information such as lines 71-74 should be moved in the section “Materials and Methods”. In other parts such as lines 69-106 is most a discussion than and introduction. Moreover at the end the aim of the work should be write in a more clear manner.

The ”Results“ should be revised because contain most of the discussion, making understanding difficult the experimental data obtained in this work. Moreover, the PSP contamination was always reported as µg STX eq. 100g tissue -1, but if other STX analogues were detected, they should be reported in text to better characterize the toxic profile of butter clams.

Also ”Materials and Methods “ should be revised because contain most of just reported in the results such as subsection 4.4. Moreover some part such as lines 294-307 should be moved in the section “Introduction”.

 The conclusions should be added, remarking the importance of the research, such as the necessity to perform official control not only on commercial harvests.

The titles of subsections and the captions of the figures should be synthesized.

The citations in the text  should be reported in the same manner.

The supplementary materials should be cited in the text and not in the captions of the figures.

Specific comments are reported in the attached file.

Comments on the Quality of English Language

The english should be improved to better understand the research.

Author Response

Comment 1. The manuscript describes the efficacy of different cleaning methods applied to butter clams, an important resource for diet of Alaskan Native communities and recreationally activities. These bivalves in fact can accumulate toxins such as saxitoxin causing Paralytic Shellfish Poisoning (PSP). The results obtained, also compared with historical data, showed that the removal of certain tissues not makes clams safe to eat. Moreover the risk associated to the consumption of 200 g of edible tissue for an average sized man and woman was evaluated, underlining that high levels of PSP contamination pose a substantial risk with moderate or severe symptoms.

The manuscript is not clear as regards the experimental results obtained, then it is also difficult to understand the novelty of the research.

Response 1. We have endeavored to sort this out more clearly. The butter clam toxicity data are from three sources.

  • Data from reports not published in primary literature (e.g.) Chambers, J. S.; Magnusson, H. W. Seasonal Variations in Toxicity of Butter Clams from Selected Alaskan Beaches; 1950. https://spo.nmfs.noaa.gov/sites/default/files/legacy-pdfs/SSRF53.pdf.

Neal, R. A. Fluctuations in the levels of paralytic shellfish toxin in four species of lamellibranch mollusks near Ketchikan, Alaska, 1963-1965. Dissertation University of Washington, Seattle, 149 pp. 1967. DOI: https://www.proquest.com/openview/7125c2f368b054c09f7110aff55ab1f2/1?pq-origsite=gscholar&cbl=18750&diss=y.

  • Comparative information from Kibler, S. R.; Litaker, R. W.; Matweyou, J. A.; Hardison, D. R.; Wright, B. A.; Tester, P. A. Paralytic shellfish poisoning toxins in butter clams (Saxidomus gigantea) from the Kodiak Archipelago, Alaska. Harmful Algae 2022, 111, 102165. DOI: 10.1016/j.hal.2021.102165
  • New data are from this study that include cleaning techniques used by members of Native Kodiak communities and the results of toxicity testing on the tissues they considered safe to eat.

Comment 2. Moreover the manner in which the paper is structured seems more a review than an article. Then a substantial revision should be done to underline the original features of the study.

Response 2. The primary argument for not structuring the manuscript as a review is the following. The data streams utilized in this paper represent new, reanalyzed, and unpublished data. The latter have not been previously published in the primary literature. The goal was to bring as many sources of information together as possible to address the question of whether or not cleaning would make clams safe to eat. In this respect we argue that this manuscript still fits better as a research article rather than a review article.  

Comment 3. The” Introduction” should be revised, some information such as lines 71-74 should be moved in the section “Materials and Methods”. In other parts such as lines 69-106 is most a discussion than and introduction. Moreover, at the end the aim of the work should be write in a more clear manner.

Response 3.The information on lines 71-74 was removed and the entire introduction revised to address the concerns about this being more like a discussion, including lines 69-106. We retained the information regarding the existing dilemma exist between what we know to be a good safety limit and how local communities perceive the risk of PSP from their direct experience consuming butter clams. Individuals in local communities know that clams deemed unsafe to eat can more often than not be consumed with no or relatively small risk which is not a deterrent to consumption. The goals of this work were to establish the basis for developing communications material establishing the random nature of encountering butter clams containing sufficient STX to cause serious illness or death, to show that no possible cleaning method can make the clams safe to eat and to put the risk in the context of “if I eat a normal sized meal, what is the chance of me getting sick”. The latter is potentially is easier for subsistence and recreational harvesters to understand than a minimal concentration-based safety limit. To better communicate these goals, the last two paragraphs now read as follows.

In this study, three different data sources of STX-eq. concentrations in butter clams were utilized to examine (1) how removing different tissues impacted the toxicity of the tissues actually consumed – i.e. can any cleaning method make the clams safe to eat, (2) the likelihood of consuming a clam capable of causing mild to serious illness, and (3) how meal size and weight of the consumer may potentially impact risk of illness. The first data set included the tissue-specific STX-eq. toxicity data from butter clams collected on Kodiak Island, Alaska by Kibler et al. [10] from 2015 to 2018. These data on the STX-eq. concentrations in various butter clam tissue were reanalyzed to estimate the degree to which removing specific tissues, or combinations of tissues, would impact the STX-eq. concentrations in the remaining portions of the clam to be eaten. The second data set is new data obtained by analyzing the STX-eq. concentrations in the edible and non-edible portions of butter clams prepared using cleaning methods practiced by Kodiak AK community members. These data are valuable because they provide specific information on methods used by local harvesters in the region and can be reported as reliable information to other communities. The third STX-eq. data set was obtained from early reports not published in primary the literature. These include clam samples collected in Southeast Alaska from 1948 to 1949 by Chambers and Magnusson [9] and 1963-1965 by Neal [13]. The toxicity of whole clams, or whole clams versus the whole clams minus the siphon, was determined using the mouse bioassay and expressed as mouse units (MU) per 100 g tissue-1 [14]. The MU toxicity values were converted to STX-eq. as described in Wekell et al. [15] for comparison with the other studies where high performance liquid chromatography methods were used to assess toxicity These data offered insight into how this method of cleaning affected the overall toxicity of the clams, plus it provided comparative information on the toxicity of butter clams collected in different regions (i.e. southeast Alaska versus the Kodiak region) where the other studies were conducted. These data can also be compared directly to the reanalyzed data from the Kibler et al. [10] study. This cross-regional comparison is important because of the evidence of differences in the toxicity of Alexandrium cells in Kodiak versus Southeast AK [16].

It should be noted that the impact of removing various tissues with regard to toxicity risk in the above studies is based on the concentration of STX-eq. in edible tissue relative to the regulatory safety limit of 80 µg STX 100 g tissue-1 for shellfish [17]. This regulation is based on the lowest concentrations of STX found in remnant meals that have been shown to cause illness divided by a safety factor of 10 [18]. Though this limit has been questioned, it is generally considered fit for the purposes of protecting public health [19]. While this approach ensures safety, it is quite conservative and may not be informative for individuals who consume butter clams regularly due to their nutritional and cultural requirements. Harvesters having eaten shellfish previously that tested above the regulatory limit with no apparent ill effects, may perceive PST-positive shellfish as posing little risk. Furthermore, toxicity estimates based on shellfish tissue concentrations do not provide an estimate of how exposure will change based on the amount of clam tissue consumed and the weight of the individual. To address this later issue, we utilized a recent modeling study by Arnich and Thébault [20] to estimate how the STX-eq. concentrations in the edible tissue and the size of the meal can impact toxicity, as well as how often harvested clams would likely caused significant illness. The resulting findings on the impacts of cleaning butter clams, from both a concentration-based and amount of toxin ingested perspective, are intended for use in developing more effective educational materials for communicating the risk of PSP to Alaskan communities.

Comment 4. The ”Results“ should be revised because contain most of the discussion, making understanding difficult the experimental data obtained in this work. Moreover, the PSP contamination was always reported as µg STX eq. 100g tissue -1, but if other STX analogues were detected, they should be reported in text to better characterize the toxic profile of butter clams.

Response 4. The reviewer is correct concerning how the variations in congeners impact toxicity. The bulk of the congener variation data was covered in the Kibler et al. 2022 paper and hence not addressed in this paper. The overall toxicity converted to STX-equivalents was sufficient address the questions being addressed int this study. To c this fact, the following line was added at the end of the last paragraph in section 2.1 – “The data on variations in the saxitoxin congener composition in individual tissues and how those differenced can affect the STX-eq. estimated is discussed in Kibler et al. 11.”

Comment 5. Also ”Materials and Methods “ should be revised because contain most of just reported in the results such as subsection 4.4. Moreover some part such as lines 294-307 should be moved in the section “Introduction”.

Response5.  With regard to lines 294-307 we respectfully disagree. Reviewer 1 requested that we simplify the introduction by removing the more technical methods information. To honor that request, and to keep the introduction more focused on the issue. To compromise we added the following two sentences to the introduction – “The toxicity of the tissues was assessed by mouse bioassay and expressed as mouse units (MU) per 100 g tissue-1 27. These MU toxicity values were converted to STX-eq. as described previously 22.

Comment 6. The conclusions should be added, remarking the importance of the research, such as the necessity to perform official control not only on commercial harvests.

Response 6. NOAA is not a regulatory agency and this type of comment is beyond the mandate of the authors to make. The manuscript has been shared with the State of Alaska health and wildlife officials and it will be up to them to proceed with any actions. We simply present the facts and let those officials who have regulatory authority make the decisions as to what policy changes are needed to address this issue.

That said, we did add a final paragraph which reads:

Given that shellfish are the third most commonly consumed traditional food among Natives in Alaska [23] and that a disproportional number of PSP victims are Alaska Natives (53%) [6], with butter clams accounting for 35% of reported PSP incidents, dissuading recreational and subsistence harvesters from consuming untested shellfish, especially butter clams, is imperative. Involving community members in testing the effectiveness of their traditional shellfish cleaning methods represents a important step in providing believable information to the population most vulnerable to PSP. The results on cleaning efficacy, the potential risk of consuming a highly toxic clam, and the amount of toxin potentially consumed in an average sized meal from this study can also be used to develop other outreach and education materials regarding the risks of consuming untested butter clams.

Comment 7. The titles of subsections and the captions of the figures should be synthesized.

Response 7. We are unsure of the request the reviewer is making here. Without additional detail we cannot address this issue beyond making the title changes suggested by the reviewer below.

Comment 8. The citations in the text should be reported in the same manner.

Response 8. We have examined the citations and endeavored to make them consistent.

Comment 9. The supplementary materials should be cited in the text and not in the captions of the figures.

Response 9. Our logic was to cite where the underlying data for each of the graphics was located in association with the corresponding graph. However, per the reviewers request they have been moved into the main body of the manuscript.

Comment 10. Specific comments are reported below:

Line 2 - Please replace “Saxitoxin” with PSP toxins”.

Response 10. The requested change has been made.

Comment 11. Line 8, 46- Please replace “(STX)” con “(STXs)”.

Response 11. In those instances where “STX” was used to represent saxitoxins generically we changed them to STXs. In those instances where we referred to saxitoxin equivalents we left the abbreviations as STX-eq. STXs-eq. in those contexts would not be appropriate because the toxicity of the various congeners present have been converted to saxitoxin (a single congener) equivalents.

Comment 12. Line 11- Please add “that “ before cleaning.

Response 12. Though we did not add the word that, we changed the sentence to read as follows: “This study tested the efficacy of cleaning methods used by harvesters on Kodiak Island, Alaska.

Comment 13. Line 16- Please add “to” before 1949.

Response 13. The hyphen was replaced by the word to.

Comment 14. Line 17- Please add “ containing a level “ before > 900.

Response 14. Changed to “Meals containing >900 µg STX-equivalents” which still keeps the abstract at 200 words.

Comment 15. Line 23- Please replace “subsistence” with “saxitoxin”.

Response 15. Subsistence was indicated to convey subsistence harvesting. The keyword has been changed to subsistence harvesting

Comment 16. Line 28 - Please replace “, Alaska” with (Alaska).

Response 16. The requested change was made.

Comment 17. Lines 43-45- Please rephase as follows : “ It is caused by saxitoxins, potent neurotoxins produced by phytoplankton species, such as Alexandrium catenella, that can bloom seasonally in cool temperate waters “.

Response 17. As requested, the sentence was revised and now reads: This illness is caused by saxitoxins (STXs), potent neurotoxins produced by phytoplankton species, such as Alexandrium catenella, that can bloom seasonally in cool temperate waters.

Comment 18. Line 48- Please replace “clams” with “bivalves”.

Response 18. In this context bivalves is not the correct term to use. 35% of all PSP cases come from eating butter clams, which can hold onto STXs for up to 2 years. The section currently reads as follows: “STXs accumulate in shellfish, with over a third (35%) of reported paralytic shellfish poisoning (PSP) illnesses in Alaska caused by consuming contaminated butter clams that may retain STXs for two years or more [5,6]”

Comment 19. Line 54- Please explain acronym for “US”.

Response 19. Now reads “United States Food and Drug Administration”.

Comment 20. Lines 52-57- Please add information about analogues of saxitoxin if they were detected.

Response 20. The various analogues are listed in the papers referenced. We have added the following sentence in the results section to address this issue at the end of section 2.1 – “The data on variations in the saxitoxin congener composition in individual tissues and how those differences can affect the STX-eq. estimates is discussed in Kibler et al. [10].”

Comment 21. Line 67- Please add a brief description on official monitoring on commercial harvests in Alaska.

Response 21. The following sentence was added: “Only commercially harvested shellfish intended for sale outside the state are tested for STXs.”

Comment 22. Lines 142-143, 202- Please add “%” after the numbers.

Response 22. Thank you for catching the omission. % has been added where noted.

Comment 23. Figure1- Please replace the graph title as follows: “ STX concentration in whole butter clams and edible tissues “.

Response 23. The requested change has been made.

Comment 24. Figure1, 2, 3, 4 - Please replace x-axis title as follows: “ Saxitoxin (µg STX-eq. 100g tissue -1).

Response 24. The requested change has been made.

Comment 25. Figure 2- Please move figure 2 before table 1 because is cited first in the text. In the caption were cited two studies but the data seems to refer to one, please verify.

Response The confusing phrase was deleted.

Comment 26. Table 1- Please clarify the correspondence with the data reported in the table 1 and the description in the text.

Response 26. To make the relationship clearer, the table legend now reads as follows: Table 1. The percentage reduction in overall toxicity in whole clams compared to edible tissue calculated from the data shown in Fig. 2.

Comment 27. Line 186- Please replace “Supplementary Table 2” with “Supplementary Table 3”.

Response 27. The change has been made as requested.

Comment 28. Lines 187-188- Please verify this information because in the Figure 1 December and January data was not showed.

Response 28.Great catch by the reviewer. Changed to Fig. 2.

Comment 29. Figure 4- Please report the STX concentrations also in samples before the cleaning to better understand the reduced toxicity applying the methods 1 and 2.

Response 29.  The black line (whole clam) indicates the concentration in the butter clams prior to cleaning.  It is intermediate between the concentrations in the non-edible tissues containing higher STX concentrations and the cleaned tissues which have a lower concentration.

Comment 30. Table 2- Please move the caption on the top of table.

Response 30. The legend was moved above the table as requested.

Comment 31. Figure 5- Please x-axis title as follows : “ Saxitoxin (µg STX-eq.).

Response 31.  The axis title was changed as requested

Comment 32. Line 238- Please add “ a contamination“ before > 200.

Response 32. Now reads: They showed that when whole clams contained a contamination >200 µg STX-eq. 100 g tissue-1 the cleaning methods rarely produced concentrations in edible tissues below the 80 µg STX-eq. 100 g tissue-1 advisory level.

Comment 33. Line 240- Please remove “had”.

Response 33. Deleted as requested.

Comment 34. Lines 285-288- Please add some information about frequency, period and sites of sampling and the total numbers of samples collected.

Response 34. The requested information has been added.

Comment 35. Line 292, 349- Please add the detector coupled to HPLC.

Response 35. PSP toxins were quantified using Agilent 1100 (Santa Clara, California, USA) or Waters Aquity Arc (Milford, Massachusetts, USA) HPLC systems equipped with fluorescence detection and 5 µm C18 columns (150×4.6 mm, Phenomenex, Inc., Torrance, California, USA). The Agilent fluorescence detector was Agilent 1100 G1321A FLD and the Waters detector was a 2475 FLR detector. Fluorescence excitation was set to 340 nm and emission to 395 nm. The mobile phase (A) consisted of 0.1 M ammonium formate, adjusted to pH 6 ± 0.1 with 0.1 M acetic acid; while mobile phase (B) consisted of 0.1 M ammonium formate with 5% acetonitrile, also adjusted to pH 6 ± 0.1 with 0.1 M acetic acid. The mobile phase was delivered by an Agilent 1200 series LC at a flow rate of 2 mL/min. The LC gradient was as follows: 0–5% mobile phase B in the first 5 min, 5–70% B for the next 4 min, hold at 70% B for 1 min, and back to 100% A over the next 2 min. The 100% A was held for an additional 2 min to allow for column equilibration before subsequent sample injections [29].

Comment 36. Line 316- Please replace the value in “rpm” with the corresponding one in “g”.

Response 36. The requested g force information was added.

Comment 37. Line 321- Please add the model of detectors and the gradient used for HPLC elution.

Response 37. The requested information was included in the section Line 292, 349 section above.

Comment 38. Line 323- Please explain acronym for “neoSTX”.

Response 38. Text was changed to “neosaxitoxin (neoSTX),..” to clarify.

Comment 39. Line 340- Please remove the details related to the linear regression.

Response 39. This information was deleted as requested.

Comment 40. Line 341-342. Please clarify the use of standard pre and post analysis.

Response 40. This paragraph containing the information in question was deleted.

Comment 41. Line 343- Please add the LOQ of the method.

Response This paragraph containing the information in question was deleted.

Comment 42. Table 3- Please clarify which tissues are considered edible and which are not since in the three studies they are defined in different ways.

Response 42. The Table 3 legend was expanded as follows to address the reviewer’s comment.

Table 3. There are various methods exist for cleaning whole butter clams by removing specific tissues prior to consumption. The tissues that are removed are categorized as non-edible, while the tissues that remain are considered edible. This table details the different ways in which tissues can be partitioned into retained edible tissue(s) and discarded non-edible tissue(s).”

Comment 43. Figure A1- Please report in the caption the reference 11.

Response 43. The Kibler et al. reference was added as requested.

Comment 44. Table S1- Please synthesize the header of columns.

Response 44. We did not understand the specific change in the column heading. In response, we have turned the labels vertically and have standardized the headings.

Round 2

Reviewer 1 Report

Comments and Suggestions for Authors

The revised paper is much improved.  

Line 18 and elsewhere.  You say 'tested for toxicity' but you are measuring concentrations of toxins.  Line 112 you say "HPLC was used to assess toxicity" which is incorrect.

Edible by definition is fit to be eaten, safe for consumption so I think it is confusing to talk about edible and non-edible portions of the shellfish in terms of what bits you are testing.  If the edible portion is toxic then it is non-edible.  Is there some other term that you can use to talk about the portions without implying safety?

Line 128.  Regarding the origin of the regulatory limit.   From my understanding this was set very early on and I haven't found mention of it originating from STX concentrations in meals.  The reference given specifies the limit but I couldn't find any information of how it was determined.  If you have a reference specifying how the limit was set then please add it or else remove lines 128-130.

Line 189.  I think it should read 'migrate to other tissues' not 'migrate in'

Line 332. Alexandrium is not required

The format of reference 18 needs correction.

Reference 18 and reference 30 are the same.

Author Response

Comment 1. Line 18 and elsewhere. You say 'tested for toxicity' but you are measuring concentrations of toxins.  Line 112 you say "HPLC was used to assess toxicity" which is incorrect.

Response 1 Response 1: The reviewer is correct. The following lines have been changed to address this issue

Line 16 - … that were tested for STX content

Line 63 - … this information about the differential toxin content of various butter clam tissues

Lines 69-70 … resulting in no potential reduction in toxin content

Lines 75-76    how removing different clam tissues impacts STX concentrations

Lines 93-106   The third STX-eq. data set was obtained from early reports that were not published in the primary literature. These reports include clam samples collected in Southeast Alaska from 1948 to 1949 by Chambers and Magnusson [9] and from 1963-1965 by Neal [13]. The STX concentration in whole clams, or whole clams versus whole clams minus the siphon, was determined using the mouse bioassay and expressed as mouse units (MU) per 100 g tissue-1 [14]. These MU values were converted to STX-eq. as described in Wekell et al. [15] for comparison with other studies where high performance liquid chromatography methods were used to assess STX concentrations. These data provided insight into how cleaning methods affected the overall toxin content of clams and offered comparative information on the toxin content of butter clams collected in Southeast Alaska versus the Kodiak region. This cross-regional comparison is important because there is preliminary evidence of toxin content differences in Alexandrium cells from Kodiak versus Southeast AK [16]. Additionally, these data can directly compared to the reanalyzed data from the Kibler et al. [10] study.

Line 107   It should be noted that the impact of removing various tissues on toxin content

Line 116   Furthermore, tissue concentrations do not fully convey to the local

Lines 126-127   How removing the viscera, the siphon black tip, or the entire siphon during cleaning affects the STX content of Alaskan butter clams

Line 151   In this case, the average toxin content in the whole clam on a 100 g basis

Line 154   every type of cleaning method failed to reduce STX concentrations below the

Line 168    The impact of removing the entire siphon on STX concentrations in butter clams collected from Southeast Alaska

Lines 173-174  overall body weight, it contributed substantially to the overall toxin content in the clam.

Line 183   Toxin concentrations were assessed using mouse bioassay and converted to STX eq. as

Line 186  The percentage reduction in overall STX-eq. concentrations in whole clams compared to

Line 188  The seasonal STX concentration patterns

Line 191   assess the STX concentrations in whole clams and the effectiveness of different cleaning

Line 195   less to the overall STX-eq. concentrations due to the absence of A. catenella

Line 198   butter clams safe to eat, a time series of whole clam STX concentration data was obtained

Line 202. that high STX-eq. concentrations can persist through the winter months

Line 204   Time series of total clam STX concentration data reported by (A) Chambers and

Line 213   reduced STX-eq. concentration in the edible tissue on average

Line 217   average the reduction in STX-eq. concentrations for Method 1 was 61.4 ± 20.4% (N=13)

Line 255   Chambers & Magnuson [9], for example, reported no correlation between STX content

Line 260   cleaning can actually increase the toxin content of edible tissues

Line 261   This is due to the removal of tissue containing lower STX-eq. concentrations

Line 262   leaving only tissues containing higher STX-eq. concentration compared to the whole clam.

Line 273   STX-eq. concentration observed were due to whether the local clam populations had

Line 414   The second study utilized STX concentration data presented in a 1954 report by

Line 417   The STX-eq. concentration in these tissues was assessed

Line 422- 423   The STX-eq. concentrations in the various tissues was assessed by mouse bioassay

Lines 449-450   samples were collected and the percent reduction in STX-eq. concentrations was calculated

Line 459-460 The studies on how different cleaning techniques might impact butter clam STX-eq. concentrations were not conducted during the colder months 

Line 463   clam STX-eq. concentrations from February 1948 to September 1949

Lines 466-467   The data for the two different time series were plotted separately to demonstrate how STX-eq. concentrations can vary seasonally

Comment 2. Edible by definition is fit to be eaten, safe for consumption so I think it is confusing to talk about edible and non-edible portions of the shellfish in terms of what bits you are testing.  If the edible portion is toxic then it is non-edible.  Is there some other term that you can use to talk about the portions without implying safety?

Response 2  We believe the edible terms non-edible tissue are valid, but understand your concerns. However, the terms edible and non-edible tissues are commonly used in manuscripts with regard to shellfish toxicity paper and their meaning is broadly understood. Rather than coming up with new terms which would require changes to all the graphics and much of the manuscript we have endeavored to make it clear that edible and non-edible tissues are what are refer to those tissues which the consumer considers tissues they would eat versus they do not eat. The additional general assumption in most communities is that the edible tissues are less toxic than the non-edible tissues. To clarify the meaning of these terms we have edited lines 64-85 to read as follows:

“This information on the varying toxin content in different butter clam tissues was shared with many coastal communities. In some of these communities it has become common practice to remove the intestinal tract, gills and black tip of the siphon before consuming the rest of the clam. Over time, a common misconception has arisen among many subsistence and recreational harvesters that by removing these tissues the remaining parts of the clam are safe to eat [10]. In other cases, clams are not cleaned at all, particularly when they are small, resulting in no reduction in toxin content. Since the State of Alaska does not test recreational or subsistence harvested shellfish, some Native communities have established their own testing procedures [11,12]. Only commercially harvested shellfish intended for interstate commerce are regularly tested for STXs.

In this study, three different datasets regarding STX-eq. concentrations in butter clams were examined to better understand (1) how removing different clam tissues impacts STX concentrations, (2) the likelihood of consuming a clam capable of causing mild to serious illness, and (3) how meal size and weight of the consumer may potentially impact the risk of illness. The first dataset consisted of a reanalysis of the tissue-specific STX-eq. concentrations measured in butter clams collected on Kodiak Island, Alaska from 2015 to 2018 by Kibler et al. [10]. We know STXs are not evenly distributed throughout each tissue [10] and these data made it possible to calculate how removing specific tissues, or combinations of tissues, would most impact the STX-eq. concentrations in the remaining portions of the clam to be consumed. Hereafter, we will refer to any combination of tissues retained for consumption as the “edible” portion of the clam and the discarded tissues as the “non-edible” portion.”

Comment 3 Line 128.  Regarding the origin of the regulatory limit.   From my understanding this was set very early on and I haven't found mention of it originating from STX concentrations in meals.  The reference given specifies the limit but I couldn't find any information of how it was determined.  If you have a reference specifying how the limit was set then please add it or else remove lines 128-130.

You were correct to ask for clarification. We have consequently revised the section regarding the origin of the 80 ug limit based on the following two references. The relevant sections relative regarding the development of the safety limit have been added for clarification.

Wekell, J.C.; Hurst, J.; Lefebvre, K. The origin of the regulatory limits for PSP and ASP toxins in shellfish. Journal of Shellfish Research 2004, 23 927–930, doi:https://www.researchgate.net/publication/285809374_The_origin_of_the_regulatory_limits_for_PSP_and_ASP_toxins_in_shellfish

From their paper - “How the specific 80 g 100 g− regulatory level was arrived at is open to some conjecture and the details are now, after over 60 years, probably lost to history. Prior to the establishment of the acid extraction method (only briefly described in Sommer &Meyer 1937) as the standard for regulatory purposes, California instituted quarantine measures when 2 mg of dried alcoholic ex-tract contained 2 MU (Medcof et al. 1947). Tests at the Laboratory of Hygiene have shown this 2 mg to be equivalent to a toxicity of400 MU per 100 g of whole meats, which is 80 g 100 g−1 in today’s parlance (Medcof et al. 1947). Canada also adopted the California standard of 400 MU as the quarantine level (Medcof etal. 1947). “

Medcof et al. (1947) also examined epidemiologic records from eastern Canada in the mid 1940s and found that in some cases a dose of 1,000 MU produced mild symptoms of PSP. Using the current conversion factor, 1,000 MU equals approximately 200 g of STX equivalents. because the lowest dose reported for illness is200 g, and assuming that 100 g of shellfish meats might represent a reasonable quantity for an adult to consume, the lowest illness-producing concentration in shellfish would be 200 g STX 100g−1. Based on these estimates and using a ×10 safety margin, the regulatory level could be set at 20 g 100 g−1. However, this is well below the mouse bioassay detection limit. Therefore, the 80 g 100 g−1 level was probably derived as a compromise based on the detection limit of the mouse bioassay (roughly 40 g 100 g−1) and yet still safely removed from the minimal toxicity of 200 g 100 g−1 observed in the early studies.”

EFSA. European Food Safety Authority. Marine biotoxins in shellfish – saxitoxin group. EFSA Journal 2009, 7, 1019

This reference provides a very detailed analysis of the background behind the determination of what is know about measurements of STX toxcity and the establishment of regulator limits using a variety of analytical methods. Page 14 includes the following informaiton.

“Section 3. Regulatory status For the control of the STX-group toxins in the European Union (EU), Commission Regulation (EC) No 853/20044 , provides details in section VII: “Live bivalve molluscs”, chapters II and IV. Chapter II: “Hygiene requirements for the production and harvesting of live bivalve molluscs. A. Requirements for production areas” states: “Food business operators may place live molluscs collected from production areas on the market for direct human consumption only, if they meet the requirements of chapter IV”. Chapter IV: “Hygiene requirements for purification and dispatch centres. A. Requirements for purification centres” states: “Food business operators purifying live bivalve molluscs must ensure compliance with the following requirements: They must not contain marine biotoxins in total quantities (measured in the whole body or any part edible separately) that exceed the following limits: for paralytic shellfish poison (PSP): 800 micrograms per kilogram [= 80 ug /100 g tissue]. This limit corresponds with most limits established in countries outside the EU, although these are often expressed differently: as μg STX equivalents/100 g. In this opinion the CONTAM Panel adopted this figure as being expressed as µg STX equivalents/kg shellfish meat.”

Finch, S.C.; Webb, N.G.; Boundy, M.J.; Harwood, D.T.; Munday, J.S.; Sprosen, J.M.; Somchit, C.; Broadhurst, R.B. A sub-acute dosing study of saxitoxin and tetrodotoxin mixtures in mice suggests that the current paralytic shellfish toxin regulatory limit is fit for purpose. Toxins (Basel) 2023, 15, 437, doi:10.3390/toxins15070437.

This article looks at the STX regulatory limits in more depth and what additional studies might be changed particularly when exposure to multiple toxins occurs. The abstract follows:

“Paralytic shellfish poisoning is a worldwide problem induced by shellfish contaminated with paralytic shellfish toxins. To protect human health, a regulatory limit for these toxins in shellfish flesh has been adopted by many countries. In a recent study, mice were dosed with saxitoxin and tetrodotoxin mixtures daily for 28 days showing toxicity at low concentrations, which appeared to be at odds with other work. To further investigate this reported toxicity, we dosed groups of mice with saxitoxin and tetrodotoxin mixtures daily for 21 days. In contrast to the previous study, no effects on mouse bodyweight, food consumption, heart rate, blood pressure, grip strength, blood chemistry or hematology were observed. Furthermore, no histological findings were associated with dosing in this trial. The dose rates in this study were 2.6, 3.8 and 4.9 times greater, respectively, than the highest dose of the previous study. As rapid mortality in three out of five mice was observed in the previous study, the deaths are likely to be due to the methodology used rather than the shellfish toxins. To convert animal data to that used in a human risk assessment, a 100-fold safety factor is required. After applying this safety factor, the dose rates used in the current study were 3.5, 5.0 and 6.5 times greater, respectively, than the acute reference dose for each toxin type set by the European Union. Furthermore, it has previously been proposed that tetrodotoxin be included in the paralytic shellfish poisoning suite of toxins. If this were done, the highest dose rate used in this study would be 13 times the acute reference dose. This study suggests that the previous 28-day trial was flawed and that the current paralytic shellfish toxin regulatory limit is fit for purpose. An additional study, feeding mice a diet laced with the test compounds at higher concentrations than those of the current experiment, would be required to comment on whether the current paralytic shellfish toxin regulatory limit should

To better reflect the derivation of the safety limit, lines 105-109 were changed to read as follows:

“It should be noted that the impact of removing various tissues on toxicity risk in the above studies is based on the concentration of STX-eq. in edible tissue relative to the regulatory safety limit of 80 µg STX 100 g tissue-1 for shellfish [15,17]. Though the specificity of this limit has been questioned, it is generally considered suitable for protecting public health [18].

  1. Wekell, J.C.; Hurst, J.; Lefebvre, K. The origin of the regulatory limits for PSP and ASP toxins in shellfish. Journal of Shellfish Research 2004, 23 927–930, doi:https://www.researchgate.net/publication/285809374_The_origin_of_the_regulatory_limits_for_PSP_and_ASP_toxins_in_shellfish.
  2. EFSA. European Food Safety Authority. Marine biotoxins in shellfish – saxitoxin group. EFSA Journal 2009, 7, 1019, doi:https://doi.org/10.2903/j.efsa.2009.1019.
  3. Finch, S.C.; Webb, N.G.; Boundy, M.J.; Harwood, D.T.; Munday, J.S.; Sprosen, J.M.; Somchit, C.; Broadhurst, R.B. A sub-acute dosing study of saxitoxin and tetrodotoxin mixtures in mice suggests that the current paralytic shellfish toxin regulatory limit is fit for purpose. Toxins (Basel) 2023, 15, 437, doi:10.3390/toxins15070437.”

Comment 4. Line 189.  I think it should read 'migrate to other tissues' not 'migrate in'

Response 4.   Changes to migrate to as requested

Comment 5. Line 332. Alexandrium is not required

Response 5. Changed to A. catenella

Comment 6. The format of reference 18 needs correction.

Response 6.  This reference has been corrected.

Comment 7. Reference 18 and reference 30 are the same.

The duplicate has been removed

Reviewer 2 Report

Comments and Suggestions for Authors

The manuscript describes the efficacy of different cleaning methods applied to butter clams, an important resource for diet of Alaskan Native communities and recreationally activities. These bivalves in fact can accumulate toxins such as saxitoxin causing Paralytic Shellfish Poisoning (PSP). The results obtained, also compared with historical data, showed that the removal of certain tissues not makes clams safe to eat. Moreover the risk associated to the consumption of 200 g of edible tissue for an average sized man and woman was evaluated, underlining that high levels of PSP contamination pose a substantial risk with moderate or severe symptoms.

The manuscript is surely more clear as regards the aim and the novelty of the study reported. In particular, it was emphasized the purpose of educating the local communities about risk of PSP.

However remains difficult to understand the results obtained, due to different sources of data (new experimental data and data reanalyzed of previous reports), then this aspect could be better clarify.

As already suggested in the previous revision, materials and methods should be revised because contain most of just reported in the results such as subsection 4.4. The conclusions should be added, remarking the importance of this research for Alaskan Native Communities.

As already suggested in the previous revision, the titles of subsections and the captions of the figures should be synthesized. 

Specific comments are reported in the attached file.

Comments on the Quality of English Language

The english should be improved.

Author Response

Comment 1. The manuscript describes the efficacy of different cleaning methods applied to butter clams, an important resource for diet of Alaskan Native communities and recreationally activities. These bivalves in fact can accumulate toxins such as saxitoxin causing Paralytic Shellfish Poisoning (PSP). The results obtained, also compared with historical data, showed that the removal of certain tissues not makes clams safe to eat. Moreover, the risk associated to the consumption of 200 g of edible tissue for an average sized man and woman was evaluated, underlining that high levels of PSP contamination pose a substantial risk with moderate or severe symptoms.

The revised manuscript is surely more clear as regards the aim and the novelty of the study reported. In particular, it was emphasized the purpose of educating the local communities about risk of PSP. However remains difficult to understand the results obtained, due to different sources of data (new experimental data and data reanalyzed of previous reports), then this aspect could be better clarify.

Response 1. The goal of the manuscript was to collate both historical and current data to reinforce the concept that untested shellfish are unsafe to eat. We agree that the data sets are indeed different in nature and do not all fit into the standard experimental format. However, we believe that the historical data in the gray literature are highly relevant and confirm the results of the more recent work as well as providing historical perspective on the current butter clam harvesting practices in Alaska. Together, each of the data sets reinforces why it is unsafe to eat untested butter clams. To try and better explain how these different datasets fit together to address the central safety question, we have expanded sections on line 64-108. That section now reads:

This information regarding the varying toxin content in different butter clam tissues was shared with many coastal communities. In some of these communities it has become common practice to remove the intestinal tract, gills and black tip of the siphon before consuming the rest of the clam. Over time, a common misconception has arisen among many subsistence and recreational harvesters that by removing these tissues the remaining parts of the clam are safe to eat [10]. In other cases, clams are not cleaned at all, particularly when they are small, resulting in no reduction in toxin content. Since the State of Alaska does not test recreational or subsistence harvested shellfish, some Native communities have established their own testing procedures [11,12]. Only commercially harvested shellfish intended for interstate commerce are regularly tested for STXs.

In this study, three different current or historical datasets regarding saxitoxin-equivalent (STX-eq.) concentrations in butter clams were examined to more fully understand (1) how removing different clam tissues impacts STX concentrations, (2) the likelihood of consuming a clam capable of causing mild to serious illness, and (3) how meal size and weight of the consumer may potentially impact the risk of illness. The first dataset consisted of a reanalysis of the tissue-specific STX-eq. concentrations measured in butter clams collected on Kodiak Island, Alaska from 2015 to 2018 by Kibler et al. [10]. We know STXs are not evenly distributed throughout each tissue [10] and these data made it possible to calculate for the first time how all the practical ways of removing specific tissues, or combinations of tissues, would impact the STX-eq. concentrations in the remaining portions of the clam to be consumed. For simplicity, we will hereafter refer to any combination of tissues retained for consumption as the “edible” portion of the clam and the discarded tissues as the “non-edible” portion.

The second data set was obtained by analyzing the concentrations of STX-eq. in butter clams that were prepared using cleaning methods commonly practiced in Kodiak AK communities. The clams were dissected into what was considered edible (body and part of the lower siphon) and non-edible (viscera, gills, black tip of the siphon) portions. Traditionally, it has been assumed that the edible portion of the clam was less toxic than the discarded non-edible tissues. The resulting data provided valuable information on methods used by local harvesters that can be shared with other communities.

The third combined data set consisted of unpublished, historical STX-eq. data from butter clam samples collected in Southeast Alaska from 1948 to 1949 by Chambers and Magnusson [9] and from 1963-1965 by Neal [13]. Both studies collected time series data on STX concentrations in whole clams, while the Chambers and Magnusson study additionally included STX concentration in whole clams minus the siphon. These data provided valuable information including: (a) Comparative data on the toxin content of butter clams collected in Southeast Alaska compared to the Kodiak region. This comparison is important due to preliminary evidence of toxin content differences in A. catenella cells between Kodiak and Southeast AK [14]. (b) Changes in toxin concentration from siphon removal that can be directly compared to one of the cleaning methods evaluated using the Kibler et al. dataset [10], (c) Publication of the original data that underpins our understanding of how cleaning may impact toxicity. (d) Information on the extent to which toxin concentrations may decrease in winter which is relevant for understanding whether cleaning clams may reduce toxicity more in the winter compare to the summer months when toxic A. catenella blooms occur.

Comment 2. As already suggested in the previous revision, materials and methods should be revised because contain most of just reported in the results such as subsection 4.4.

Response 2.   In response to this suggestion, we have endeavored to remove extraneous information from the materials and methods sections and particularly section 4.4. These changes were made through this section are noted in the track changes section and not printed here.

Comment 3. The conclusions should be added, remarking the importance of this research for Alaskan Native Communities

Response 3. A conclusion section was added to address suggested change and now reads as follows:

Conclusions.

Given that shellfish are the third most commonly consumed traditional food among Natives in Alaska [21] and that a disproportional number of PSP victims are Alaska Natives (53%) [6], with butter clams accounting for 35% of reported PSP incidents, dissuading recreational and subsistence harvesters from consuming untested shellfish, especially butter clams, is imperative. Our results unequivocally show that no cleaning method renders untested butter clams safe to eat. The findings further indicate that though the probability of severe illness from consuming cleaned clams is not common it can occur with no means of reliably predicting when and where highly toxic clams will be harvested. Mild to moderate illness from consuming cleaned clams is relatively common. Involving community members in testing the effectiveness of their traditional shellfish cleaning methods as was done in this study represents an important step in providing believable information to the population most vulnerable to PSP. The results on cleaning efficacy, the potential risk of consuming a highly toxic clam, and the amount of toxin potentially consumed in an average sized meal from this study can also be used to develop other outreach and education materials regarding the risks of consuming untested butter clams.

Comment 4. As already suggested in the previous revision, the titles of subsections and the captions of the figures should be synthesized. For example:

Comment 4 responses.

We thank the reviewer for providing more specific guidance on the title changes they considered necessary.

  • 2.1 Effects on toxicity of Alaskan butter clams of different cleaning methods

Response: now reads - The effects of various cleaning methods on the toxin concentration in Alaskan butter clams

  • 2.2 Impact on toxicity of Southeast Alaska butter clams of siphon removing

Response: Now reads - The impact of siphon removal on toxin concentration in Southeast Alaska butter clams

  • 2.2.2 Comparison of two cleaning methods practiced by Alaskan Native communities

Response: used the suggested title

  • 2.2.3 Risk associated with consuming meal contaminated by PSP

Response: now reads - Risk associated with consuming meal contaminated by saxitoxins

Comment 5

  • Figure 1. STX-eq. concentrations for whole butter clams and different types of edible tissues, collected from June 2015 to July 2018. The yellow dashed line indicates the regulatory limit(80 µg STX eq. 100 g tissue -1).

Response 5 – Legend now reads

Figure 1. The µg STX-equivalents 100 g tissue-1 for whole butter clams and the different types of edible tissues listed in Table 3. The yellow dashed line indicates the regulatory limit (80 µg STX eq. 100 g tissue-1 for safe consumption of shellfish). N=39.

Comment 6.

  • Figure 2. Time series of STX-eq. concentrations in whole butter clams, as well as edible (body and gut) and non-edible (all the siphon tissue) tissues collected in Southeast Alaska from 1948-1949 [9]. The black dashed line represents the regulatory limit (80 µg STX eq. 100 g tissue -1).

Response 6. Legend now reads as follows:

Figure 2. Time series of STX-eq. concentrations in whole butter clams, as well as edible (body and gut) and non-edible (entire siphon) tissues collected in Southeast Alaska from 1948-1949 [9]. The black dashed line represents the regulatory limit. The black dashed line represents the regulatory limit (80 µg STX eq. 100 g tissue-1).

Comment 7

  • Figure 3. Time series of total clam toxicity data reported by (A) Chambers and Magnusson (1948-1949) [9] and (B) by Neal (1963-1965) [13]. The blue shading on the graphs indicates the period from November to March when water temperatures are cooler. The black dashed line indicates the regulatory limit (80 µg STX eq. 100 g tissue -1).

Response 7. The figure legend now reads as follows. We would prefer to include the n numbers in the legend:

Figure 3. Time series of total clam STX concentration data reported by (A) Chambers and (1948-1949; N=250) [9] and (B) by Neal (1963-1965; N=123) [13]. The blue shading on the graphs indicates the period from November to March when water temperatures are cooler. The black dashed line indicates the regulatory limit (80 µg STX eq. 100 g tissue-1).

Comment 8

  • Figure 4. Time series of the STX-eq. concentrations in the edible and non-edible butter clam tissues prepared following (A) Method 1 and (B) Method 2. Each month three replicate batches were collected on the same day (April to July 2018 for Method 1, April to October 2018 for Method 2). The individual replicates are plotted separately to show inter batch variability. The black dashed line indicates the regulatory limit (80 µg STX eq. 100 g tissue-1 ).

Response 8

The figure legend now reads as follows:

Figure 4. Time series of the STX-eq. concentrations in the edible and non-edible butter clam tissues prepared following (A) Method 1 and (B) Method 2. Each month three replicate batches were collected on the same day (April to July 2018 for Method 1, April to October 2018 for Method 2). The three individual replicates collected each month are plotted separately to show inter batch variability. The black dashed line indicates the regulatory limit (80 µg STX eq. 100 g tissue-1; N=21).

Comment 9

  • Figure 5. STX-eq. concentrations that would be ingested if 200g of edible tissue was consumed. The concentration data used to determine the quantity of STX ingested were from the 2015-2018 Kodiak study shown in Fig. 1.

Response 9. The figure legend now reads:

Figure 5. STX-eq. concentrations that would be ingested if 200g of edible tissue was consumed. The concentration data used to determine the quantity of STX ingested were from the 2015-2018 Kodiak study shown in Fig. 1. N=39.

Specific comments are reported below:

Comment 10

Line 29 - Please replace “accumilate”with accumulate”.

Response 10. Replaces as requested

Comment 11

Line 52- Please remove “paralytic shellfish poisoning”and the brackets.

Response 11: Removed as requested

Comment 12

Line 58- Please replace the “end point”with“comma” after vomiting.

Response 12: replaced as requested

Comment 13

Lines 81-82- Please add references to legislation concerning the control on commercially harvested shellfish.

Response: Added reference

Comment 14

Line 83- Please explain acronym for “eq.”

Response 14: now explained on lines 73 – 74 which read as follows:

In this study, three different current or historical datasets regarding saxitoxin-equivalent (STX-eq.) concentrations

Comment 15

Line 100-Please explain acronym for “AK”.

Response 15: Now define don line 30 - The butter clam (Saxidomus gigantea), found from the Aleutian Islands (Alaska, AK)

Comment 16

Line 103- Please remove “STX-eq”.

Response 16: Removed as requested

Comment 17

Lines 162-166- Please rephrase, the sentence is too long then not clear.

Response 17:  Still a long sentence but now broken up using a,b,c,d designations. Sentence reads as follows:

The overall percent reduction in STX concentration relative to whole clams for the various edible tissues was as follows: (a) whole clam minus gut, 6.8 ± 20.1%; (b) whole clam minus gut and siphon black tip, 7.9 ± 8.5%; (c) whole clam minus entire siphon, 9.6 ± 17.3%; and (d) clam body only, 15.4 ± 17.8%.

Comment 18

Line 220- Please rephase as follows “ from May 1G48 to January 1G4G “.

Response 18: Read correctly in the current copy I have. Not sure why it rendered as letter G instead of a 9.

Comment 19

Line 240- Please replace “2.2.2”with “2.2.1”. Check also the following subsections.

Response: made the requested change and checked the numbering

Comment 20

Line 258- Please replace “STX-equivalents”with “ STX contamination”.

Response 20: Replaces as requested

Comment 21

Line 276- Please replace “saxitoxin”with“STX”.

Response 22: Replacement made as requested

Comment 22

Line 277- Please replace “ from Method 1”with“by Method 1” and“(N=13)”with“(N=12)”.

Response 22: now reads - On average the reduction in toxicity by Method 1 (N=12) was 61.4 ± 20.4% relative to whole clams and for Method 2 it was 45.2 ± 23.3% (n=21) (Supplementary Table S4).

Comment 23

Line 278- Please replace “n” with “N”.

Response 23: Replaced as requested

Comment 24

Figure 4- Please use the same colour of dashed line for A and B graphs.

Response: Changed as requested. Both lines are now black.

Comment 25

Line 293- Please replace “in”with“by” before Arnich.

Response 25: replaced as requested.

Comment 26

Line 298- Please rephase as follows “ in the 2015-2018 Kodiak study“.

Response 26:  changed as requested

Comment 27

Line 316-317 - Please add “from” before 200 and 500.

Response 27: added from as requested

Comment 28

Lines 318-319- Please rephase, the sentence is not clear.

Response 28: the sentence now reads - Because of the high degree of variability in STX-eq. concentrations in individual clams, determinations regarding the safety of butter clams are based on assaying batches of 12 homogenized clams.

Comment 29

Line 399- Please add “coupled to fluorescence detection (FLD)“ after(HPLC).

Response 28: As requested the sentence now reads HPLC systems coupled to fluorescence detection (FLD) and 5 µm C18 columns (150×4.6 mm, Phenomenex, Inc., Torrance, California, USA).

Comment 30

Line 405- Please remove mouse units and the brackets.

Response 30: the mouse units and brackets were removed as requested

Comment 31

Line 408- Please remove “saxitoxin equivalents”and the brackets.

Response 31: Removed as requested

Comment 32

Line 418- Please remove high performance liquid chromatography and the brackets.

Response 32: Removed as requested

Comment 33

Line 421- Please remove the space.

Response 33: space was removed

Comment 34

Lines 435-436- Please rephase as follows “ The flow rate was set at 2 mL/min”.

Response 34: The sentences now read as follow - The mobile phase was delivered by an Agilent 1200 series LC. The flow rate was set at 2 mL min-1.

Comment 35

Lines 443-444- Please replace “GTX1/4, GTX2/3, C1/C2”with “GTX1,4, GTX2,3, C1,C2”.

Response:

Comment 36

Line 446- Please remove toxicity equivalence factors and the brackets.

Response 36: Now reads as follows - Isomers GTX 1,4, GTX 2,3, and C1,C2 are not separated using

Comment 37

Lines 454-460- I think that it is important to include some information regarding the performance of the method such as LOQ.

Response: 37 – added the following sentence - The LOQ data for this method are reported in Turner et al. [27]

Comment 38

Line 466- Please add“-FLD” after HPLC.

Response 38:-FLD was added after HPLC

Comment 39

Line 478- Please remove “exist”.

Response 39: Removed as suggested

Comment 40

Line 491- Please replace “tissue” with “tissues”.

Response: s added as requested.